


# Using Machine Learning to Predict Optimal Electromagnetic Induction Instrument Configurations for Characterizing the Root Zone

Kim Madsen van't Veen[a], Ty Paul Andrew Ferré[b], Bo Vangsø Iversen[a], Christen Duus Børgesen[a]

[a] Department of Agroecology, Aarhus University, Blichers Allé 20, 8830 Tjele, Denmark
[b] Department of Hydrology and Atmospheric Science, University of Arizona, Tucson, AZ 85721.

**Correspondence:** Kim M. van't Veen (km@agro.au.dk)

**Abstract.** Electromagnetic induction (EMI) is used widely for hydrological and other environmental studies. The apparent electrical conductivity ($EC_a$), which can be mapped efficiently with EMI, correlates with a variety of important soil attributes. EMI instruments exist with several configurations of coil spacing, orientation, and height. There are general, rule-of-thumb guides to choose an optimal instrument configuration for a specific survey. The goal of this study was to use

machine learning (ML) to improve this design optimization task. In this investigation, we used machine learning as an efficient tool for interpolating among the results of many forward model runs. Specifically, we generated an ensemble of 100,000 EMI forward models representing the responses of many EMI configurations to a range of three-layer subsurface models. We split the results into training and testing subsets and trained a decision tree (DT) with gradient boosting (GB) to predict the subsurface properties (layer thicknesses and EC values). We further examined the value of prior knowledge that

could limit the ranges of some of the soil model parameters. We made use of the intrinsic feature importance measures of machine learning algorithms to identify optimal EMI designs for specific subsurface parameters. The optimal designs identified using this approach agreed with those that are generally recognized as optimal by informed experts for standard survey goals, giving confidence in the ML-based approach. The approach also offered insight that would be difficult if not impossible to offer based on rule-of-thumb optimization. We contend that such ML-informed design approaches could be

applied broadly to other survey design challenges.

## 1 Introduction

Water movement through the vadose zone is often controlled by the near surface hydrogeologic structure. In the simplest sense, this is often represented as a small number of horizontal layers, such as is often related to soil formation processes





leading to distinct soil layers. The hydrogeologic structure places critical controls on processes ranging from infiltration to
percolation to root water uptake to recharge, thereby playing a critical role in most hydrologic systems (Winter et al., 1998;
Nimmo, 2009). The need to describe this shallow hydrogeologic structure has been a major driver in the development and
adoption of hydrogeophysical methods (Binley et al., 2015).

Electromagnetic induction (EMI) is a non-contact method to measure the apparent electrical conductivity ($EC_a$) of the
shallow subsurface. The $EC_a$ is an integration of the electrical conductivity of all layers in the subsurface. EMI works when a
transmitter coil produces an electromagnetic field that induces secondary currents in the subsurface soils. The combined
current is measured with a receiver coil (Nabighian and Macnae, 1991). The strength of the measured field is used to
estimate the $EC_a$ within the sample volume of the measurement (Doolittle and Brevik, 2014). EMI instruments differ in the
orientations of their coils: some use transmitter and receiver coils that have their long axis horizontal with respect to the
ground surface (HCP), others orient both coils vertically (VCP), and some use one horizontal and one vertical coil in a
perpendicular arrangement (PRP). In addition, instruments differ in the separation of the coils, with larger separations used
to measure to greater depth. Finally, an operator can choose different instrument heights above ground, which also impacts
the spatial sensitivity of the measurement in the subsurface. We refer to the collective choices of coil orientation, separation,
and height above ground as the instrument configuration.

For several decades, EMI instruments have been used to gather measurements of $EC_a$ of the soil. The $EC_a$ of soil is positively
correlated with salinity, water content, and clay content (Doolittle and Brevik, 2014). As a result, $EC_a$ is a meaningful, but
complex, aggregate measure of soil properties (Palacky, 2011). Because the EMI method is non-contact, it is reasonably fast
and inexpensive compared to direct soil sampling, resulting in a frequent use in agriculture (McCutcheon et al., 2006;
Daccache et al, 2015; Adhikari and Hartemink, 2017), soil mapping (James et al., 2003; Cockx et al., 2009; Heil and
Schmidhalter, 2012; Reyes et al., 2018), and archaeological investigations (Saey et al., 2013; De Smedt et al., 2014; Saey et
al., 2015; Christiansen et al., 2016). In addition to the challenges introduced by $EC_a$ being sensitive to multiple soil
properties, quantitative interpretation of EMI measurements is complicated by the complex averaging of the local soil EC
within the instrument's sample volume. (Note that we use the term EC to refer to the actual bulk electrical conductivity of a
soil, which may vary within the measurement volume of the instrument, and $EC_a$ to refer to the average EC that is measured
from EMI instrument responses.) More challenging still, the spatial sensitivity (or spatial weighting) of the EC depends on
the instrument configuration (McNeill, 1980). Finally, in some cases, the spatial sensitivity may depend on the absolute
value and spatial distribution of the EC (Callegary et al., 2012). In this investigation, we make the common assumption that
the spatial sensitivity only depends on the instrument configuration, but this dependence could be considered using more
complete forward models of EMI response. The spatial averaging of EMI is not an issue if the medium is electrically
homogeneous. However, most soils have some structure – at a minimum, agricultural soils display horizontal layering with a
distinct uppermost layer (the Ap horizon). Therefore, optimal design of an EMI configuration should select the orientation,
separation, and height of the coils to locate the instrument sensitivity in the subsurface to best determine the subsurface



properties. Developers of EMI instruments have long recommended using different configurations to measure layered ECa

values, leading to simple rules of thumb such as using shorter coil separations for shallow mapping and larger separations for deeper investigations. However, these basic guides become more difficult if the objective is to determine subsurface properties in a non-homogeneous medium, even a simple layered case. For these conditions, a nonexpert user is often advised to use different coil orientations with the same separation or some combination of orientation, separation, and height. But little specific guidance is offered. Furthermore, there is no way for a user to consider the possible impact of prior

knowledge (e.g. bounds on the expected depth of the topmost layer) in the survey design. Commercially available EMI instruments for relatively shallow applications offer a wide range of designs based on differences in the three instrument characteristics. This makes it difficult for t users without theoretical background in geophysics to make an informed choice regarding the preferred instrument and configuration.

There are several published efforts to optimize the design of geophysical surveys (e.g. Furman et al., 2007; Khodja et al.,

2010; Song et al., 2016) Applying these design optimization approaches to EMI would require that the responses of many configurations be computed for multiple soil models. Each survey design includes multiple measurements at each location, each with a different configuration, that jointly provide the most useful information for inferring specific, user-identified subsurface properties. That is, a user is faced with the question of which *combination* of configurations is optimal given their measurement priorities and, ideally, incorporating any applicable constraints that they may have regarding the subsurface

conditions. Any method that requires formal inversion of each proposed combination of configurations is computationally intractable for most users.

Machine Learning (ML) describes a wide range of regression algorithms used for pattern recognition. ML has grown in popularity and is now used regularly within and beyond science. The simplest ML tools are based on Decision Trees (DT), which are supervised ML techniques that perform classification or regression by sequential categorization based on

observations. For our application, each $EC_a$ measurement made with a different EMI configuration represents a feature in ML parlance. By training DTs on many examples, they can be used to efficiently predict outcomes based on observations without formal, model-based inversion. DTs are computationally inexpensive, but they can have limited predictive skill (Hastie et al., 2001). To improve their performance, DTs are often augmented by ensemble learning methods such as bagging (Breiman, 1996) and boosting (Friedman, 2001). For our application, we found that gradient boosting (GB) offered

improved performance without adding unreasonable additional computational effort. Feature importance key ability of DTs (with and without GB), which is a functions that quantify the importance of each feature for making the predictions of interest. We make use of this ability importance for EMI survey design optimization.

We used DT with GB as an efficient approach to EMI measurement design optimization. Specifically, we ran many forward models of EMI response for a range of three-layer subsurface conditions (varying each layer thickness and EC). We then

tested the ability of DT with GB to infer the correct value of each subsurface property given the $EC_a$ that would be measured with all the EMI configurations. We used the feature importance capabilities of DT with GB to identify which observed $EC_a$ values were most informative for the inference and eliminated all insensitive configurations. This allows us to find the





optimal instrument configurations for each subsurface parameter without having to do multiple inverse models, one for each possible combination of observations for each parameter. To examine the impact of independent knowledge of any of the

subsurface properties, we then repeated this analysis for a subset of the soil models that met a given restriction, such as only those that had a thin upper layer or a high EC middle layer.

The engine for our analysis is EMagPy (Mclachlan et al., 2020), a recently published open-source code that offers ready access to forward and inverse modeling for a wide range of users. For this analysis, we only made use of the forward modeling ability of EMagPy. We then used the EMagPy output as the input for a python code that implemented the DT with

GB analyses and produced the figures to guide EMI survey design. The ultimate goal was to develop an approach to measurement optimization that would be accessible to a wide range of users, with the hope that a similar approach could be developed for other measurement network design problems. The specific objective of this investigation was to present an approach to select sets of EMI configurations that are optimal given the specific survey goals and any independent knowledge of the subsurface electrical properties.

## 2 Theory

### 2.1 Depth sensitivity of EMI instruments

If the subsurface is electrically homogeneous within the sample volume of the instrument, then the EMI instrument response ($EC_a$) can be related directly to the EC of the subsurface. It is more common, especially on agricultural soils that are not subject to net percolation, that the EC varies with depth due to soil layering, irrigation, or near-surface accumulation of salts.

For these conditions, multiple measurements, made using different coil spacing and separations, can be interpreted simultaneously to infer the EC profile. This requires a model of the depth sensitivity of the EMI measurement.

The simplest, most widely used depth sensitivity model is the Cumulative Sensitivity (CS) model of McNeill (1980). This analytical solution describes the contribution from the soils below any given depth to the measured $EC_a$. The model only strictly applies under low induction number conditions and the response depends only on the depth, coil separation length,

and coil configuration with no regard for the subsurface EC distribution. Taking $z$ to be the depth divided by coil separation and adding the instrument height above the surface to the depth, the CS response factors, $R$, of the three coil configurations are:

$$R_{VCP}(z) = \sqrt{(4z^2 + 1)} - 2z, \tag{1}$$

$$R_{HCP}(z) = \frac{1}{\sqrt{(4z^2+1)}}, \tag{2}$$

$$R_{PRP}(z) = 1 - \frac{2z}{\sqrt{(4z^2+1)}}, \tag{3}$$





The contribution from a single layer is given by the EC of the layer weighted by the CS response factor. The contributions from all layers are summed to define the total response ($EC_a$). Imagine a subsurface with two distinct layers with a top layer with a conductivity of $EC_1$ and thickness of $t_1$ and the lower layer of infinite thickness and $EC_2$. For the specific condition where the thickness of the top layer is equal to the coil spacing, z, the $EC_a$ from an HCP would be:

$$EC_a = EC_1 * [1 - R_{HCP}(z)] + EC_2 * R_{HCP}(z), \tag{4}$$

More complete solutions have been developed that remove or reduce the restrictions of McNeil's solution (Monteiro Santos 2004; Auken et al., 2015; Saey et al., 2016). EMagPy (McLachlan et al., 2020) offers the user the opportunity to use several models and makes them readily available to a wide audience, even users with no background in EMI modelling.

## 3 Materials and Methods

In this study, we describe a specific EMI instrument configuration based on the three coil orientations horizontal (HCP), vertical (VCP), perpendicular (PRP), coil separation (in m), and instrument height (in m). For example, a configuration that uses coils that are horizontal to the surface with a separation of 1 m and an instrument height of 0.3 m would be named: hcp_1.0_0.3. The EC of any layer is an actual electrical property of that specific medium and it is referred to as EC followed by the layer name. For example, the EC of the A-layer is referred to as ECA. Likewise, the thickness of any layer is denoted by Thick followed by the layer name. Thus, the thickness of the A-layer is denoted as ThickA. All symbols and abbreviations can be found in Appendix A.

### 3.1 Generating the model ensemble

We consider a three-layer soil profile, which is common for agricultural soils with distinctly developed A-, B- and C-layers characterizing changes in the physical, chemical and biological characteristics with depth (Fig. 1). Electrical properties are assumed to be constant horizontally within the sample volume of the instrument. The subsurface properties (three EC values and two thicknesses) were varied independently (Table 1), forming a large set of subsurface conditions. Then, the $EC_a$ was calculated for many EMI instrument configurations using EMagPy (Mclachlan et al., 2020) version 1.1.0.



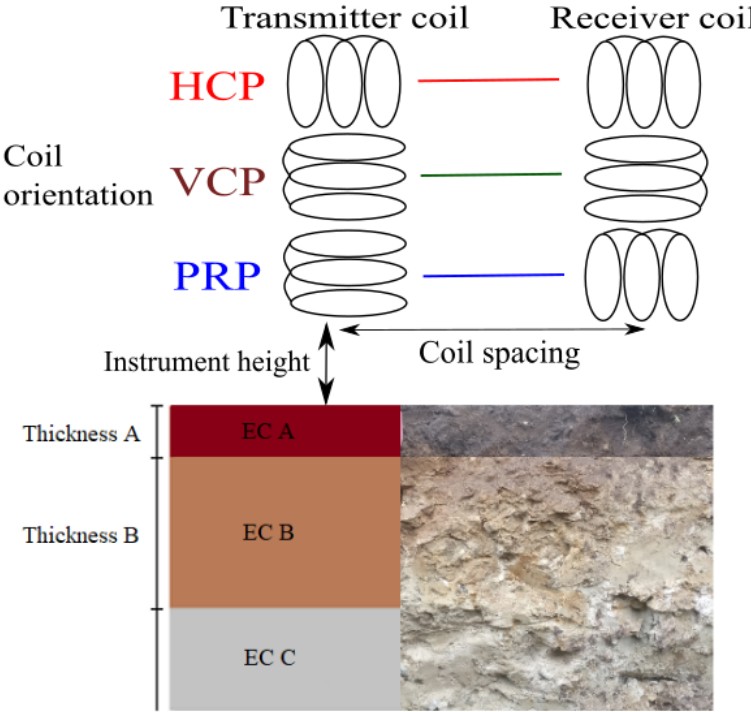

**Figure 1: Three layered soil (A-, B-, and C-layer) with variable electrical conductivities (EC). Also showing the schematic of an EMI instrument situated on the surface. The HCP and VCP has the receiver coil is in the same horizontal plane as the transmitter coil. The PRP have the receiver coil in the plane perpendicular to the transmitter coil.**

Each of the five soil parameters had ten possible values, which created 100,000 different EC soil profiles. The ranges of EC used in the forward model were chosen to represent a wide spectrum of soil types, water contents, and salinities. The lowest EC represents a dry sandy soil and the highest EC represent an agricultural soil with a combination of high clay, salinity, or water content (Triantafilis and Lesch, 2005; Robinson et al., 2008; Harvey and Morgan, 2009). The ranges of soil layer thicknesses ranged from thin (0.05 m) to relatively thick (2.0 m) for agricultural sites. Each of the three coil orientations was modelled for three different coil separations and three different instrument heights, all of which are typical for field applications of EMI with commercially available instruments. In total, the EMagPy code was run 2.7 million times to form the ensemble of results covering the soils and instrument configurations. Note that all analyses were repeated for the Andrade (2016) EMI model. The findings were not significantly different, so the results are presented for the simpler, more widely used McNeil model.





**Table 1: Adjustable parameters used in the forward model to generate the ensemble and values used for each of the combinations that constitute the soil profiles.**

| Subsurface parameters | | | | |
|---|---|---|---|---|
| ECA | ThickA | ECB | ThickB | ECC |
| [mS/m] | [m] | [mS/m] | [m] | [mS/m] |
| 1 | 0.05 | 1 | 0.1 | 1 |
| 12 | 0.21 | 12 | 0.3 | 12 |
| 23 | 0.37 | 23 | 0.5 | 23 |
| 34 | 0.53 | 34 | 0.7 | 34 |
| 45 | 0.69 | 45 | 0.9 | 45 |
| 56 | 0.86 | 56 | 1.1 | 56 |
| 67 | 1.02 | 67 | 1.4 | 67 |
| 78 | 1.18 | 78 | 1.6 | 78 |
| 89 | 1.34 | 89 | 1.8 | 89 |
| 100 | 1.5 | 100 | 2.0 | 100 |
| Instrument parameters | | | | |
| Height | | Coil spacing | | Coil orientation |
| m | | | | |
| 0.1 | | 1.0 | | Vertical |
| 0.3 | | 2.5 | | Horizontal |
| 0.5 | | 4.0 | | Perpendicular |

175

### 3.2 Analyzing the EMI model results and feature importance with a gradient boosted decision tree

#### 3.2.1 Decision tree models

Decision tree is a machine learning method that performs regression or classification practicing on subset of the full data set called training data. A training data set consists of n samples $(x_1, y_1)$, $(x_2, y_2)$, …, $(x_n, y_n)$, where x is the inputs (features) and

180 y is the response (target). The aim is to estimate a function F(x) that connect the features with the targets in a way that minimizes the loss function (Friedman, 2001):

$$L\big(y, F(x)\big) = \sum_{i=1}^{n} \frac{1}{2} [y_i - F(x_i)]^2, \tag{5}$$



The features in our dataset consists of values of modelled $EC_a$ from various instrument configurations and the targets are the five adjustable subsurface parameters. The tree is built by splitting the values of the features in the training data into two groups. The optimal split minimizes the sum of squared residuals between the value of the targets and the average value of all target within each group. The two new groups are split into additional two groups each (Hastie et al., 2001). This process continues creating a structure like an upside-down real-world tree with a root node at the top, from which non-terminal nodes (branches) will be at every split, and terminal nodes (leaves) at every end point. To avoid overfitting the growth of the tree is limited by introducing a maximum depth of the tree and a minimum number of data samples required to create a leaf by splitting a non-terminal node.

### 3.2.2 Gradient boosting algorithm

The GB algorithm (Friedman et al., 2001; Mason et al., 1999) takes the training data set and the chosen loss function to make an initial estimate $F_0(x)$ as a starting point. When the loss function is defined by eq. 5 the initial estimate $F_0(x)$ becomes the average of the inputs $x_1$, $x_2$, …, $x_n$. The residual $r_{im}$ between the initial estimate calculated by $F_0(x)$ and the true value of the targets are calculated for i=1, 2, …, n:

$$r_{im} - \left[ \frac{\partial L(y_i, F(x_i))}{\partial F(x_i)} \right]_{F(x) = F_{m-1}(x)}, \tag{6}$$

Equation 6. is the gradient from which the algorithm named, and the residual $r_{im}$ are named pseudo-residuals. Then a decision tree model is made from the features to predict the pseudo-residuals from eq. 6. The decision tree model output is scaled by a learning rate v to reduce variance of the prediction. The scaled output is added to $F_0(x)$ to create a new function $F_m(x)$ for decision tree m for i=1, 2, …, n:

$$F_m(x) = F_{m-1}(x) + v \sum_{j=1}^{J_m} \gamma_{jm} I(x \in R_{jm}), \tag{7}$$

Where $J_m$ is the total number of leaves in the terminal region $R_{jm}$ in decision tree model m. The new function $F_m(x)$ is used to calculate a new set of pseudo residuals. The process of making a new decision tree model $F_m(x)$ and adding the scaled output to the existing function $F_{m-1}(x)$ is repeated until the reduction in pseudo residuals with each added tree becomes insignificant or a specified number of trees M has been created.

Feature importance is an indicator of how valuable each of the included features is in the context of the final decision with GB. The relative importance $\hat{I}_j^2$ of any feature is proportional to the number of times it is used to make splits weighted by the square of its improvement to the goodness of fit for the model at each split (Friedman and Meulman, 2003):



$$\hat{I}_j^2(T) = \sum_{t=1}^{J-1} \hat{\iota}_t^2 1(v_t = j), \tag{8}$$


which sums over the non-terminal nodes J-1 in the tree T and the squared residual $\hat{\iota}_t^2$ attributed to the split of each node t with $v_t$ as the target variable being split at each node (Friedman, 2001). Since boosting generates multiple trees the relative importance is averaged over all trees. The importance is normalized over all features so that the sum of the feature importance values equals one where a higher value indicates a greater effect on the targets.


Gradient boosting (Elith et al., 2008; Friedman, 2001) was used for all analyses. A separate boosted tree was trained to predict each of the five subsurface parameters. The EMI model ensemble was split into training and testing sets, with 70% used for training and the remaining 30% used for testing, using the random sample function in python. Training and testing were repeated five times with different training/testing splits. Differences among the repeats were small, so all results were

combined for analyses. The learning rate, maximum tree depth, and minimum samples per leaf were tuned manually and the optimal values for these parameters were found to be 0.1, 10, and 2, respectively. However, the performance of the DT with GB did not vary significantly with the hyperparameter values. All other hyperparameters used the default values in the scikit-learn toolbox (Pedregosa et al., 2011).

**3.3 Assessing the value of additional information**

For our initial analyses, we considered the full range of all the subsurface electrical properties. However, in many cases, prior information is available to define one or more of these soil EC parameters or, at least, to reduce the range of plausible values for at least one of them. This prior knowledge could be in form of hard data or soft expert knowledge for a survey area. Here, we examine how reducing the uncertainty of one soil EC parameter improves the EMI-based inference of other

parameter values and whether this additional information changes the composition of the optimal EMI configurations to include in a survey.

To examine the value of additional a-priori parameter information, we perform three restriction analyses. In each case, we sequentially limit the range of one of the five subsurface EC parameters and determine the impact on the accuracy of inference of the other parameters. Recognizing that some parameters, especially EC values, can have a different impact on

EMI energy distribution if they are high or low valued, we consider four patterns of restriction:

- Centered: The four minimum and four maximum values defining the parameter ranges are eliminated.
- Skew low: The eight highest values are eliminated from the parameter range.
- Skew high: The eight lowest values are eliminated from the parameter range.
- Full range: All ten possible values of the five parameters are used in the analysis.





For each restriction analysis, we present the impact of the restriction compared to the case with no independent information and we describe any changes in the composition of the optimal EMI configuration set for each target subsurface parameter.

## 4 Results and discussion

In this section, we present the outcome from the forward modelling with EMagPy (section 4.1). We also assess the results from applying a DT with GB to output of the forward modelling. First we look at parameter identifiability and examine the

cases that lead to inaccurate predictions (section 4.2) and then we examine the feature importance output (section 4.3). We show the impact of restricting the range of ThickA on inferring ECA (section 4.4.1). Analysis described in sections 4.1 to 4.4.1 focuses on the full range of parameters and ECA, the EC of the A-layer (the shallowest layer). Finally we present the impact of piecewise applying all restriction patterns to all five subsurface parameters on the value of independent information (section 4.4.2) and the feature importance of EMI configurations (section 4.5).

### 4.1 Modelled $EC_a$ ensemble

The five soil parameters with ten different values provides us with an ensemble of 100,000 soil profiles. The three coil orientations, three coil spacings, and three instrument height sums to 27 instrument designs that are applied to each profile. Frequency distributions of the modelled $EC_a$ for each of the 27 instrument designs in all the profiles are shown in Fig. 2. The distributions are quite similar, but they do differ in detail. The distributions of modelled $EC_a$ values depend strongly on the

height or coil orientation for designs with a 1-meter coil separation (left column, Fig. 2). The variations are less pronounced for larger coil separations. There are also differences in the smoothness of the distributions: the PRP (bottom row, Fig. 2) has more distinct peaks for small separations whereas the HCP (top row, Fig. 2) has more peaks for larger separations.





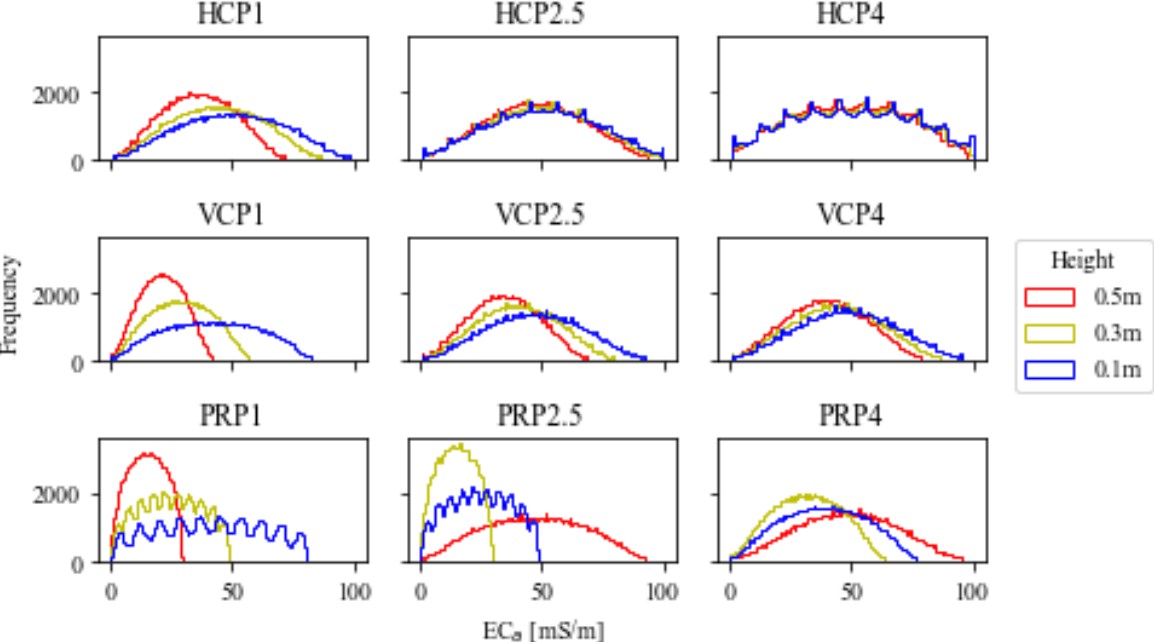

 **Figure 2: Frequency distributions of the responses from the cumulative sensitivity model for the three coil orientations: Horizontal (HCP), vertical (VCP) and perpendicular (PRP). Each panel shows the modelled $EC_a$ output from one coil orientation and -separation for three different heights. The coil orientation and -separation change respectively with the rows and columns of the nine panels.**

## 4.2 Predicting parameter values with a trained DT with GB using all observations

The first step in our analysis was to examine the ability of the trained DT with GB to predict each parameter value. That is, we use 70,000 EC profile realizations for training the DT with GB. We then provide the 27 observations for each of the remaining 30,000 EC profile realizations to the trained DT with GB and predicted ECA (the EC of the shallowest layer). To account for the brittle nature of DT methods, this procedure was repeated five times with different training/testing splits. The results of the repeated analysis were not significantly different, so they were pooled, providing 150,000 predictions upon

which the goodness of fit was determined.

The root mean squared error (RMSE) between predicted and true values of the EC of the A-layer (ECA) is shown on Fig. 3. The true values are the known ECA values used in the forward models. The results, shown as a cross-plot of points, are somewhat misleading because it is difficult to see that many points are overlapping close to the 1:1 line. Therefore, shaded areas are included to show ± one and two standard deviations about the mean predicted ECA for each true ECA value. There

are clear outliers – cases for which the trained DT with GB did not give an accurate estimate of ECA even considering all 27 EMI observations. However, the overall RMSE was 7.34 mS/m over the entire set of 150,000 test cases.



**Figure 3: The result from running the DT with GB on the entire 100000 soil types and all 27 instrument configurations five times.**
**The EC of the A-layer (ECA) is the parameter that is being predicted. The X-axis is the true value of the ECA, and the Y-axis is**
**the predicted values for ECA.**

The process shown in Fig. 3 was repeated for each of the five EC profile parameters. The RMSE for each parameter is

reported in Table 2. Because the range of values of the parameters differ, the normalized root mean square error (NRMSE) is

calculated by dividing the RMSE by the full range of the true values of the parameter. The results show that EMI is least able

to infer the layer thicknesses, with slightly better ability to infer the thickness of the A compared to the B-layer. Furthermore,

EMI produces better estimates of the shallow and deep EC values compared to the EC of the B-layer. These results fit with

expectations, given that EMI designs with very short coil separations might be sensitive to only ECA and those with very

large separations might be mostly sensitive to the EC of the deepest layer, ECC (Callegary et al., 2012; Heil and

Schmidhalter, 2015). In contrast, the layer thicknesses, ThickA and ThickB, and the EC of the middle layer, ECB, must

always be inferred based on multiple measurements.





**Table 2: The root mean square error (RMSE) between the prediction from the gradient boosted (GB) model and the testing data. The machine learning procedure was repeated with each of the five subsurface parameters as targets, thus creating five models. The RMSE is normalized by the mean value of the target to get the normalized root mean square error (NRMSE).**

| Target | ECA | ThickA | ECB | ThickB | ECC |
|--------|------|--------|------|--------|------|
| Unit | mS/m | m | mS/m | m | mS/m |
| RMSE | 7.34 | 0.29 | 18.7 | 0.49 | 1.51 |
| NRMSE | 0.07 | 0.20 | 0.19 | 0.26 | 0.02 |

### 4.2.1 Examining the conditions that led to poor estimations

From the 150,000 test cases, displayed on Fig. 3, 8,894 cases are more than one standard deviation away from the true value when predicting ECA. These cases are displayed in Fig. 3 by the blue markers that are located outside the shaded areas. The compositions of these 8,894 cases are presented as frequency distributions of their parameter values in Fig. 4. The values for ECB, ECC, and ThickB are uniformly distributed, which indicates that no specific values of ECB, ECC or ThickB lead to poor inference of ECA. In contrast, 94% of the problematic conditions have a thickness of the A-layer (ThickA) among the

three lowest values. This, again, agrees with expectations that the EC of a thin layer would be more difficult to infer accurately than that of a thicker layer using an EMI instrument. The finding is opposite for ECA; while not as pronounced, the results indicate that the poorly inferred cases tended to have higher ECA values, with 54% of the conditions having the three highest ECA values. Practically, this suggests that the method would be more likely to be successful if a user can be relatively certain that the range of ThickA does not include the lowest values examined here; that is, we would expect

improved inference of ECA for centered or high skewed restrictions of ThickA. A more successful survey, based on the ability to infer ECA, would occur if the ECA values tend to be lower. That is, a center or low skewed restriction should show better performance.



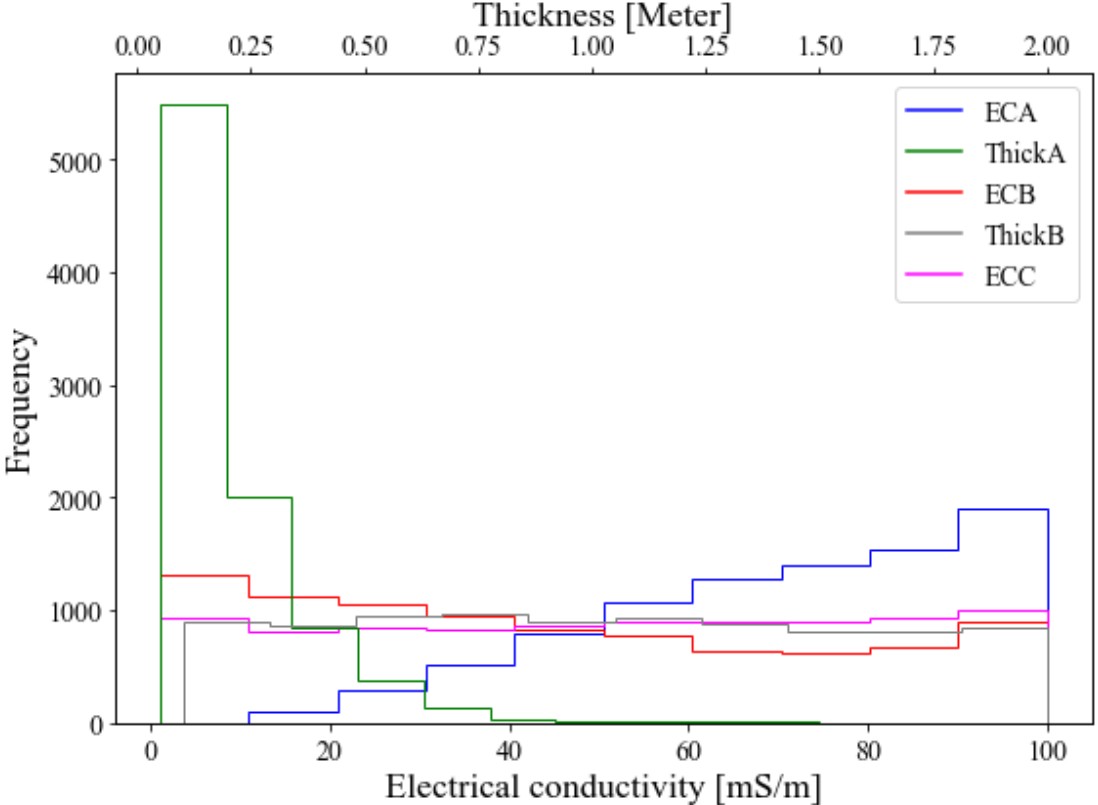

**Figure 4: The ECA was inferred for 150,000 test cases. In 8894 of the 150,000 cases the inference was more than one standard deviation away from the true value. The figure shows the distribution of five subsurface parameter values within the 8894 conditions. The top X-axis is the layer thickness, the bottom X-axis is the layer EC and the Y-axis is the frequency.**

### 4.3 Feature importance when predicting parameter values with a trained DT with GB

The preceding analysis used measurements from all 27 instrument configurations for each EC profile parameter estimation. The major focus of this investigation was to use ML tools to identify the optimal set of observations to collect, which balances performance with reduced field effort. To illustrate how the built-in feature importance of tree-based methods can be used to achieve this, consider the results shown on Fig. 5. The feature importance is shown for each of the 27 configurations; because they sum to 1 it is convenient to represent this as a pie chart. The colors and patterns that comprise the circles identify the eight most important EMI configurations for each combination of the parameters. The fraction of the cirlce covered by each color/pattern shows the relative importance of that observation. The colors indicate the coil orientation, while the shade and pattern indicate the coil distance and instrument height. The 19 least important EMI configurations are combined in "others" (white slices). From these results, it is apparent that approximately 90% of the information used to predict ECC (rightmost circle) is provided by configuration hcp_4.0_0.1. The optimal orientation and



large coil separation could have been predicted from McNeil's classic work (McNeill, 1980). However, he did not consider
the PRP orientations. The reason for the preference for a small instrument height is as apparent; it may simply be due to

further penetration of the signal to greater depth. To our knowledge, no other method, short of exhaustive comparisons of
many synthetic inverse analyses, would have been able to show that a single configuration was so clearly dominant for
inferring ECC. Similarly, almost 60% of the information used to infer ECA (leftmost circle) was provided by the
prp_1.0_0.1 configuration. The small coil separation and low instrument height fit with general expectations, but the PRP
orientation was not expected before conducting this analysis.

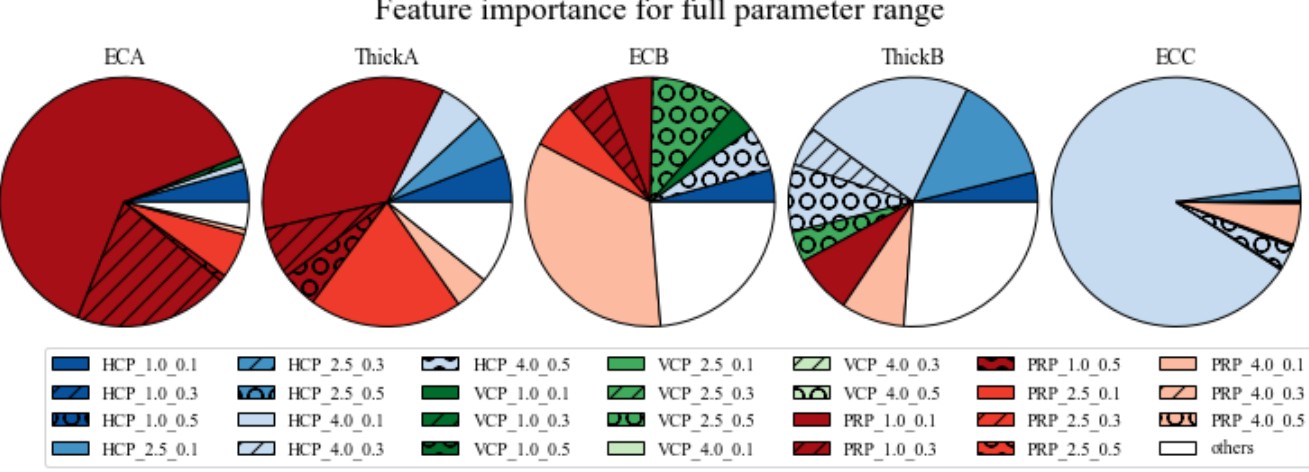

**Figure 5: Feature importance for inferring each of the five parameters from a decision tree analysis of the full parameter range.
The feature importance from all 27 configurations sum to 1. The eight most important configurations for inferring each of the five
parameters are shown with a unique color and pattern combination. The remaining 19 configurations are aggregated into the
"others" category and displayed with white.**

Taken together, the results suggest that each of the EC profile parameters relies on a relatively small number of observations.
To illustrate this, 90% of the importance, including only the highest importance observations, is provided by 3, 8, 14, 17, and
1 observation for ECA, ThickA, ECB, ThickB, and ECC, respectively (Fig. 5). Of these high importance observations, 53%
had the instrument placed at the lowest instrument height considered. Perhaps more controversially, in the context of EMI
instrument design and use, only 26% of the most informative configurations used the VCP orientation (Fig. 5). This may be

partially explained by the spatial sensitivities of the orientations (Callegary et al., 2007; Christiansen et al., 2016) which
indicates relatively high spatial sensitivity redundancy for the HCP and VCP orientations.





## 4.4 Parameter restriction analyses

### 4.4.1 Applying a skew low restriction to the thickness of layer A

One piece of information that may be available (e.g. from direct field examination) is the expected thickness of the shallow
topsoil layer (ThickA). Therefore, we begin our restriction analyses by examining the effect of improved knowledge of
ThickA on the inference of the ECA parameter. Specifically, we repeated the analysis only including models with the two
middle values of ThickA (0.69 m and 0.86 m). This reduces the ThickA parameter range to 11% of its full range and thereby
removes the cases that contains low values for ThickA. The results (Fig. 6) show stark improvement in the ability of the DT
with GB to infer ECA. A similar analysis could be repeated for any restricted range of value for any parameter or for
multiple parameters. This could be done for practical reasons – to design a site-specific survey – or for scientific reasons – to
explore which conditions are identifiable with EMI and to understand these parameter interactions.

The analysis leading to Fig. 6 is one example of the ability of the DT with GB method to consider the benefits of
independent soil property information. In this section, we expand the investigation to include all the soil electrical
parameters and three different restriction patterns.




**Figure 6: The result from running the machine learning algorithm on a subset of the ensemble where the thickness of the A-layer have been restricted. Only 20,000 soil types and all 27 instrument configurations remain in this restricted subset. The EC of the A-layer (ECA) is the parameter that is being predicted.**

### 4.4.2 Changes in parameter inference of restricted subsets

Figure 7 summarizes the impacts of providing the maximum additional information (considering only two of the ten possible

values of one parameter) on the inference of all other parameters. The y-axis on Fig. 7 is the RMSE (such as that reported on

Figure 6 for inferring ECA with ThickA restricted) normalized by the full range (max – min) of the inferred parameter. With

reference to Fig. 6, this would be reported as the RMSE divided by the range of ECA, giving a unitless value of 0.028. Each

inferred parameter is associated with a short horizontal line, which indicates the normalized RMSE without restriction of any

other parameter's range. Each symbol on Fig. 7 represents the results of an analysis like that shown on Fig. 6. There are three





symbols (triangle, dot, and square) associated with each target/restricted parameter pair for each of three restriction patterns. Consider, for example, inferring ECA. The set of three blue symbol represents the impact of restricting the range of ECA itself, the leftmost triangle represents skewed low restriction (retaining the two lowest ECA values), the middle dot is a centered restriction (ECA values 45 and 56 mS/m), and the right square represents the skewed high restriction (retaining the two highest ECA values). As expected, restricting the range of ECA, regardless of the restriction pattern, leads to a similar reduction in the normalized RMSE of ECA. Every pair of restricted/inferred parameters is represented using three symbols with the same left nudged triangle, center dot, right nudged square for the low, middle, and high skewed restrictions.

Consider another example to illustrate how Fig. 7 can be interpreted and related to Fig. 6. The three symbols dots above ECA represent the impact of restricting ThickA. The center dot corresponds exactly to Fig. 6, the centered restriction of ThickA. The left green triangle shows that there is an increase in the NRMSE for the skewed left restriction compared to the unrestricted case (horizontal line above ECA), which shows that restricting the thickness of layer A to the lowest range of values leads to lower quality inference of ECA. In other words, the shallowest layer may be too thin to be detected properly because the instrument response is an integration over a large depth compared to the now relatively thin layer thickness. This fits with previous findings (Fig. 4), which revealed that a thin ThickA makes it difficult to infer ECA. Furthermore, it agrees with our expectations that if the uppermost layer is sufficiently thick, we can choose an coil separation and orientation that is almost exclusively sensitive to the uppermost layer, essentially allowing direct measurement of ECA. Consistent with this explanation, the right green square above ECA has the lowest NRMSE. In this case, this confirms the expectation that it is easier to infer ECA accurately if the shallowest soil layer is relatively thick. Similar interpretations about the value of restricting one parameter on the ability to infer other parameters accurately can be drawn for each pair of restricted/inferred parameters, allowing users and researchers to gain valuable insight into the interaction of measurements and other independent information. In all cases, there is a reduction in the NRMSE of the inferred parameter when the parameter itself is restricted. For these cases, there are no significant differences among the three restriction patterns. In most cases, restricting the range of the inferred parameter itself showed a greater improvement than restricting any other parameter. The only clear exception was inferring ECA, which showed a greater improvement by restricting ThickA with a central or right skew.



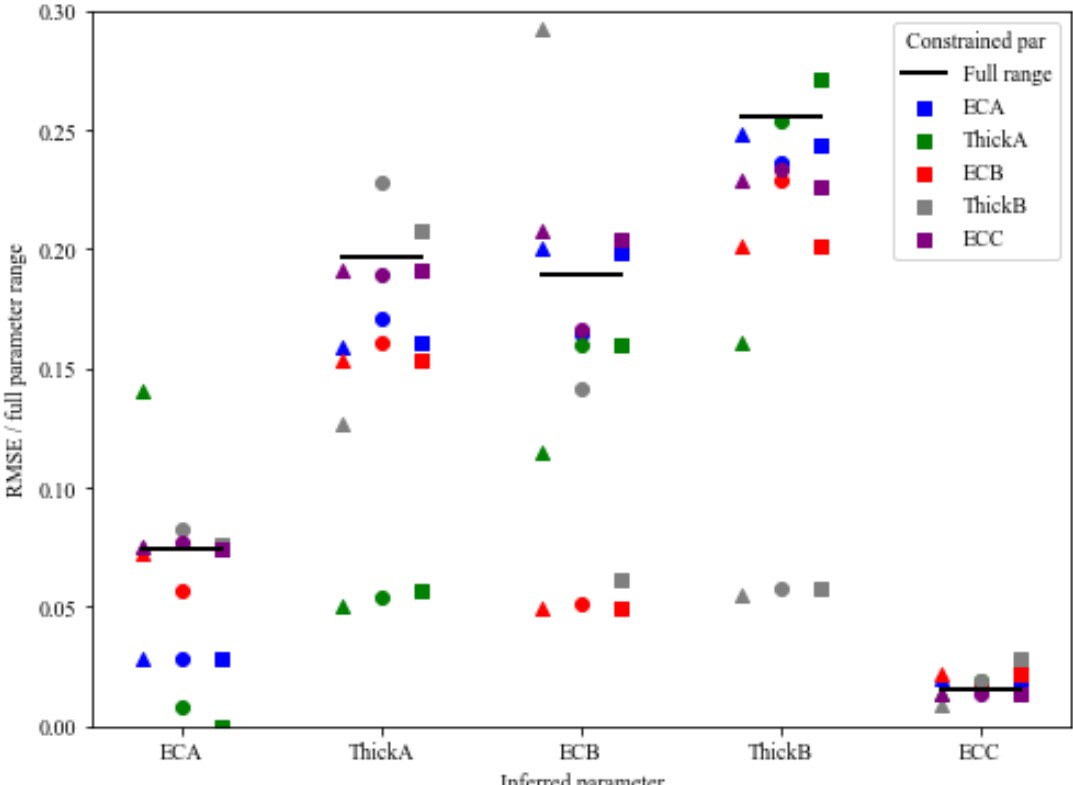

**Figure 7: The changes in inference of the five subsurface parameters (x-axis) are based on a comparison between the RMSE from restricted case divided by the range of the parameter (Y-axis). The lines show how well the parameters are predicted when all parameters are full range. The color shows which parameter that is being represented and the location and symbol represents the three restriction patterns skewed low (left nudged triangle), centered (centered dot), skewed high (right nudged square).**

In practice, Fig. 7 can be used as a guide for planning an EMI survey by helping to prioritize which information is most likely to improve the inference of any specific parameter value of interest. Consider the inferred parameter ThickB on Fig. 7. The three green symbols represent the cases where ThickA is restricted. The left triangle is the skewed low restriction that results in a reduced NRMSE compared to the full parameter range (black line). The middle dot, which is centered restriction, shows the same NRMSE as the full parameter range. The right square, which is skewed high restriction, has a higher NRMSE than the full parameter range. The changes in NRMSE between the three restrictions of ThickA show that knowledge of the ThickA confers little advantage to estimating ThickB unless it can be shown that the shallowest layer is very thin.

More generally, there are relatively few cases where the restriction of one parameter significantly improves the inference of another parameter. Beneficial restrictions include restricting ECA and ECB to infer ThickA and restricting ThickA and ECA to infer ECB. To a lesser degree restricting any other parameter when inferring ThickB offers a slight advantage. The value of ECC is already well constrained for the full parameter range, as shown by the line, and there is little advantage to restricting another parameter to infer ECC. In 24% of cases, restricting the range of one parameter led to worse inference of





another. These cases can guide a user to field conditions that lead to more challenging use of EMI, such as a very thin middle

layer making it very difficult to infer ECB. From the perspective of an experienced user of EMI surveys, most of these
general conclusions will be obvious, which helps to confirm the validity of the proposed approach. We see the value of this
analysis as providing general guidance to less experienced users and to provide more fine-tuned guidance for site-specific
conditions for those with more experience using EMI. Furthermore, the guidance provided is quantifiable rather than based
on general rules-of-thumb.

**4.5 Feature importance in restricted subsets**

The composition of the optimal EMI measurement configuration is different depending on the soil layer thicknesses and
conductivities. Fig. 8 summarizes the feature importance for the cases presented in Fig. 6, for which only two out of ten
values remain for the restricted parameter. The color and symbol patterns are the same as those used for Fig. 5. The columns
in Fig. 8 represent the five inferred parameters and the rows represent the restricted parameter. Consequently, each circle is a

pairing between one restricted and one inferred parameter. The circles are subdivided into four rings that represent the
different restriction patterns. From inside out, the rings represent the full parameter range (no parameter restriction),
centered, skew low, and skew high restriction. The feature importance of the full parameter range (centermost ring) is the
same in every row for each inferred parameter. For reference, the center ring results are identical to those presented in Fig. 5.
All 75 combinations of the five inferred/restricted parameters and the unrestricted case are shown for the three restriction

patterns on Fig. 8, allowing a user to draw general insights into the value of different configurations under a wide range of
conditions.



**Figure 8: Feature importance for the 8 most important EMI configurations for every combination of the five inferred/restricted parameters and the three patterns. Each circle is subdivided into four rings that shows, from inside out, the feature importance for full range, centered, skew low, and skew high. Each column/row represents the each of the five inferred/restricted parameters. The coil orientations are colored so that Horizontal (HCP) is blue, Vertical (VCP) is green, and Perpendicular (PRP) is red. A dark and light hue represents respectively a short and long coil distance.**

Figure 8 is somewhat information dense, so it may be useful to discuss a few cases in more detail. One of the simplest subplots to understand is the inference of ECC when restricting ECA (top right circle). The results show clearly that there is no meaningful change in the composition of the optimal set of configurations due to adding additional ECA information, regardless of the range of ECC values considered: all four concentric rings look nearly identical. Furthermore, all four rings indicate that a single configuration, HCP_4_0.1 provides the vast majority of the information needed to characterize ECC. Again, this is in general agreement with the rules of thumb provided by McNeil (1980), but it confirms these findings for all values of EC and thickness of the other layers, and it extends the findings to consider the PRP configuration. Moving down





the ECC column, note the difference when ThickB is restricted. If ThickB is skewed high (ThickB ranges between 1.8 m and 2.0 m), there is some advantage to adding the PRP_4_0.1 configuration. Our approach does not explain this choice. We suggest that it is informative to collect this additional observation to constrain the values of ECB and ThickB if the middle layer is relatively thick and that the identified configuration has a usefully different sensitivity distribution than the large HCP array placed close to the ground surface. This result could not be anticipated based on McNeil's solutions. Furthermore,

the resulting optimal configuration is almost identical if either ThickA or ThickB is restricted, when inferring ECC. Moving to the bottom of that column, the analyses show that if the value of ECC itself is limited then the composition of the optimal set changes significantly. Interestingly, regardless of the pattern of restriction (the results are the same for the outer three rings), the optimal set now includes four configurations with approximately equal importance: HCP_4_0.1; HCP_4_0.3; HCP_2.5_0.1; and PRP_4_0.1. It is further confirmation of the validity of the approach that no VCP arrays were chosen, as

would be expected based on McNeil (1980). Similarly, as expected, the larger array separations are preferred. It is surprising, however, that one of the four observations place the instrument higher above ground. We suggest that this is a good example of a result that has both immediate practical value for survey design and could point researchers to ask follow-on questions about why this combination of observations is identified as optimal.

The results for inferring ECA (leftmost column) are similar but show interesting differences. The optimal set for ECA is
relatively insensitive to the pattern of restriction of ECA. But, more than one observation is required for all cases. Whereas the optimal cases were similar for restricting ThickA and ThickB for inferring ECC, this similarity holds for restricting ECB and ThickB when inferring ECA. The pattern of restriction of ThickA has dramatic impacts on the optimal set of configurations for inferring ECA. The three other parameters (ThickA, ThickB, and ECB) show significant changes in the optimal configuration set depending upon the pattern of restriction (ring-to-ring) and upon the independent information

provided (row-to-row). There is no case for which a single configuration dominates the importance. In fact, there are many cases that would recommend more than nine configurations. For example, this likely indicates that ThickB is unlikely to be well resolved by a practical field survey. Further considerations of inferring ThickB give interesting general insights compared to rule-of-thumb suggestions. Namely, very few VCP configurations are selected. If PRP arrays are to be used, then profiling should be achieved by increasing the coil separation with the coil placed close to the ground. For HCP

configurations, profiling should be achieved by increasing the coil separation and by lifting the instrument above the ground for the largest coil separation configuration.

To summarize, taken together Fig. 7 and 8 provide a direct guide to an EMI user when designing a survey with a specific target. Figure 7 indicates whether that target can be characterized reliably given the full range of configurations considered and which additional information will improve the characterization. Figure 8 identifies the optimal set (and number) of

arrays needed for optimal characterization. Some of the conclusions would be expected based on McNeil's (1980) classic work and would be anticipated by an experienced EMI user. Other results would be difficult, if not impossible, to predict without a value-of-data analysis like that shown here. These results, in particular, could point the way to further scientific investigations to better understand the complementary information content of multiple EMI configurations. The restriction





analyses offer insight into the mutual identifiability of soil EC. Given the availability and flexibility of EMagPy (Mclachlan
et al., 2020) and the efficiency of the DT with GB algorithm, the analyses performed here could be extended to include
identification of optimal configuration sets for multiple targets (e.g. thickness and EC of the B layer). For example, placing
equal weight on all five targets, an optimal without restriction of any of their values suggests the use of: one HCP array
(hcp_4.0_0.1) and four PRP arrays (1.0_0.1, 4.0_0.1, 1.0_0.3, and 2.5_0.1). If this specific set of configurations was deemed
impractical, a user could limit the available configurations for consideration, find the optimal survey, and compare the
projected RMSE to that estimated for the overall optimal set. This information could guide a user in whether it is worthwhile
to change their instruments, or designs, or whether gathering additional information about the range of plausible parameter
values is likely to be more important for their survey goals. Finally, the general approach shown here could be extended
easily to consider multiple measurement types (e.g. combining EMI with other geophysical methods), and even dynamic
optimization of measurement networks for monitoring applications.

## 5 Conclusions

Most environmental and agricultural field investigations are conducted on relatively limited budgets. As a result, there is
usually some advantage optimizing data collection to achieve the best results with the limited time and money available.
These restrictions are one of the main reasons that electromagnetic induction (EMI) has become a popular tool for these
studies. While it is often the case that the measurements are more ambiguous than direct measurements of soil properties, the
noncontact nature of the instruments allows for much greater spatial coverage. The recent availability of EMagPy
(Mclachlan et al., 2020), allowed us to perform the large number of EMI forward models necessary to support a machine
learning (ML) examination of EMI surveys, leading to a simple but comprehensive investigation of parameter identifiability
and optimal EMI configurations. The result is an approach that can allow an EMI user with limited expertise to choose a
better set of instrument configurations given their main survey goal and knowledge of the site conditions. The same tool can
point more advanced users to areas of investigation that may improve our understanding of the prior knowledge content of
different EMI configurations. The decision tree with gradient method based on a large ensemble of instrument response
forward models, proposed here, makes novel use of the efficiency and built-in feature importance capabilities. But, the
analyses are not restricted to this relativley simple ML algorithm. More advanced ML tools could be combined with
independent feature importance analyses if required for specific monitoring applications. Similarly, while EMI forward
modeling is relatively simple and fast, given that it is based on analytical models, with sufficient computational resources
any measurement method and underlying physical process could be examined in the same way. As just one illustrative
example, an optimal combination of EMI, electrical resistivity, gravity, and monitoring well observations could be proposed
to constrain the interpretation of a pumping test performed in an unconfined, anisotropic medium by conducting forward
models of many configurations (survey locations and times, Electrical Resistivity Tomography array types, and screen
depths) for a large ensemble of plausible aquifer conditions and allowing an ML algorithm to consider all of the data and



identify the most informative observations. This opens the possibilities for exploring truly novel combinations of multimodal observations.

## 6 Appendices

### Appendix A Symbols and abbreviations

CS – Cumulative sensitivity

DT - Decision Trees

EC – Electrical conductivity

$EC_a$ – Apparent electrical conductivity

ECA/B/C – Electrical conductivity of layers A, B and C

EMI – Electromagnetic induction method

GB – Gradient boosting

HCP – Horizontal co-planar

ML – Machine learning

NRMSE – Normalized root mean square error

PRP – Perpendicular-planar

RMSE – Root mean square error

ThickA/B – Thickness of layers A and B

VCP – Vertical co-planar

### Code and data availability

The modelled EMI data and code used in this study is available on https://zenodo.org/record/4621121

### Author Contributions

KM contributed with software, visualization, interpretation, and drafted the manuscript. TF contributed with conceptualization, interpretation, supervision, and writing. BI and CB contributed with supervision and manuscript revisions.





**Competing interests**

The authors declare that they have no conflict of interest.

**Acknowledgements**

The main author was funded by a PhD scholarship from GSST, Aarhus University. This study was supported by the Innovation Fund Denmark projects: MapField—field scale mapping for targeted N regulation and management (8850-00025B), and rOpen—open landscape nitrate retention mapping (6450-00006B). This work was carried out and funded as a
part of the activities by the Aarhus University Centre for Water Technology, WATEC.

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
