# Peer review of "Using Machine Learning to Predict Optimal Electromagnetic Induction Instrument Configurations for Characterizing the Shallow Subsurface"

_Hydrology and Earth System Sciences, 2021_

## Author Comment (AC1)

**Response from the authors to the comments by anonymous referee**

We would like to thank the referee for providing constructive review and commentary.

I very much appreciate to investigate the numerous options of ML for geophysical application and novel ideas related to this issue are of particular interest for HESS. In this manuscript ML was intended to be used to improve a design optimization task for electromagnetical field mapping. The approach is interesting and especially the interpretation of the feature importance has an added value as this allows some enhanced interpretation.

We are happy that you appreciate the investigation of machine learning for geophysical application and find our approach interesting. We agree that the interpretation of feature importance is the most interesting nugget!

The manuscript holds a lot of interesting results however I suggest to rethink the focus of the manuscript. In the recent form of presenting the methods and results I cannot agree that "The result is an approach that can allow an EMI user with limited expertise to choose a better set of instrument configurations given their main survey goal and knowledge of the site conditions. (line 493/494)".

One of my concerns is that the authors formulate as their main objective to present an approach to select sets of EMI configurations that are optimal given the specific survey goals and any independent knowledge of the subsurface electrical properties - with the aim to support users with limited expertise, see line 67-74. To fulfill this aim it would be more helpful to write a practical guideline than a scientific paper. In the recent form I have doubts that the manuscript can support users with limited expertise as the figures and way of recommendation needs to be simplified.

Thank you for finding our results interesting. We agree that we should step back from the goal of making a simpler approach and redirect the focus towards the scientific value of the study. We have refocused the paper significantly based on the reviewer's recommendations and greatly appreciate their perspective. This has led to a fundamental change in the objective of the paper that we find much more compelling – again, we thank the reviewer for their insight.

Moreover the authors choose a rather arbitrary selection covering a very broad range of subsurface properties for the forward models. The chosen ECa range is rather high and from the practical point of view many field sites vary by a delta ECa not more than 20 mS/m which would cover only two classes (e.g., van Hebel 2018, McLachlan 2017, Robinet 2018, Reyes 2018).

The full ranges of the subsurface properties are supposed to cover the range of many areas. This is to simulate a scenario where the same user must survey multiple areas that not necessarily similar and we therefore consider a wide range of geology, which can have a large variation in EC (Palacky, 2011). This could apply to an investigator that is tasked with surveying multiple fields but wanting to keep the design the same for intercomparison purposes, or who is conducting a survey over a large or rather heterogeneous area.

However, our later analysis shows how a user can choose to only consider a narrow range of values if the site conditions are better defined. When we constrain the subsurface ranges in section 4.4 and 4.5 it is to illustrate that there can be a benefit to changing the instrument setup based on the specific field. Figure 1 shows the ECa measured with a horizontal coil at 2 meters separation. On this field the range of ECa values varies from 1.6 mS/m to 99.3 mS/m. While this kind of variation might not be the norm, we left in the possibility that it can occur. In addition, the approach could be constrained to consider high resolution

within a narrower range of EC values to give a user insight into how finely EC could be constrained with EM instruments.

[Figure]

Figure 1 Raw ECa measurements from the horizontal coil with 2 meter separation in a dualem21 instrument. The field is located at coordinates 56°07'40.3"N 9°51'45.0"E in the central Jutland, Denmark.

Given the option of EMagPy it seems to me more convenient, even for an unexperienced user, to run a forward model with several instrument configurations (HCP, VCP, PRP and coil distances) for the specific application with some prior knowledge of texture, salinity etc..

The purpose of this approach is to reduce the bias that comes from the suggested approach. How does a user decide on which small set of configurations to consider? How do they quantitatively compare the likely success of these proposed configurations? Our idea is to provide a simple, objective approach that can explore many possible configurations – including some that may not be in popular use. Furthermore, even considering only a few configurations, the user would have to consider multiple combinations of these configurations, which quickly becomes impractical. If we then include a sensitivity based on existing knowledge the number of simulations can become huge and interpretation requires more effort than most investigators will commit. (Perhaps this is one reason that so few pre-survey analyses are conducted to optimize data collection.) We choose an illustrative example of using layer EC and thickness as prior knowledge. But any information could be used to constrain the range of cases that is considered by the machine learning.

Moreover I see a big challenge for unexperienced users to understand the dynamic aspects of the depth sensitivity of EMI depending on the subsurface EC distribution. In this manuscript this aspects was excluded as stated in line 58-59/line 120. I can understand to keep the situation in a first attempt simple in terms of using McNeill model, however I would strongly avoid to make decision on measurement configurations without keeping this aspect in mind.

This is a good point. Fortunately, because EMagPy includes forward models that consider (to some degree) the impacts of conductivity structure on the EMI response, this would be a trivial extension. If the conditions warranted the added effort (i.e., the LIN assumptions are clearly violated), then the user could implement an even more complete forward model within the ML structure shown here; the only cost would be the forward model run time. Our choice to use McNeil was based on two things. First, McNeil is still the most widely used model for interpreting EMI data – we contend that the data collection should be chosen with consideration of how the collected data will be analyzed. Second, we wanted to make the connection between ML recommendations and underlying concepts. For a broader audience, we felt that these discussions would be clearer if based on the relatively simple cases for which McNeil applies. (If the reviewer is interested, we discuss why a more complex forward model actually provides even greater advantages for our proposed approach compared to traditional inverse model approaches to data worth analysis in response to the other reviewer's general comments.)

My suggestion would be either

- to focus on a very practical guide for users based on forward modelling that not only includes the instruments configuration but also EC of the subsurface and including a real world example to transfer knowledge into practice

- or to focus on the scientific value of the study and rather present and discuss your approach (and its advantages) compared to existing approaches/forward modelling having more room for a structured discussion (e.g. Table 2, Figure 4 and Figure 7) and advancing the way of presenting the results (Fig 7, 8). Especially for the results in chapter 4.4 I do not see the added value clearly.

We appreciate the reviewer's advice. We have significantly refocused the paper on the scientific value – how ML can provide an objective approach to assessing the likely information content of a wide range of possible measurement sets. However, we have maintained some extension of the work into practical implications because we feel that EMI is, ultimately, a highly applied method more so than a research-grade instrument.

We see the value of section 4.4 analysis as providing a quantifiable way of assesing how well an EMI survey will fare depending on the goals and and field conditions of the survey. Rather than depending on a rule of thumb (see below). The change in NRMSE creates a measure of how idientifiable a parameter is. Instead of suggesting that thin layers are hard to detect we can quantify how much harder they are to detect and at what thickness it becomes impractical to use EMI.

We now have explicitly defined the general rule of thumb in the introduction:

"The depth of investigation (DOI) of EMI instruments is both in the scientific literature (Saey et al., 2009a; Saey et al., 2009b; Saey et al., 2012; De Smedt et al., 2014; Doolittle & Brevik, 2014; Adamchuk et al., 2015) and by the manufacturers (Dualem Inc., Canada n.d.) often estimated to be at the depth the has 70% of the cumulative response. There is a relationship between depth sensitivity of the instrument response and coil

spacing and position. Therefore 70% cumulative response rule is in practice frequently converted to a rule of thumb that states larger coil spacings and HCP should be used for deeper investigations while short spacing and VCP/PRP should be used for shallow investigation (Acworth, 1999; Beamish, 2011; Cockx et al., 2009; K Heil & Schmidhalter, 2015; Kurt Heil & Schmidhalter, 2019). While this rule of thumb is not wrong, the terms shallow and deep are subjective and will have different meaning depending on whether it is a hydrogeologist, archeologist, agronomist or a geophysicist who applies the terms. It also fails to make any distinction to the differences between using the VCP or PRP coil orientations."

We edited the aim to:

"One of the challenges of both scientific and environmental investigations is to determine the optimal data to acquire. Data, which is often used to provide structural information to a model or constrain model parameterization. Measurement optimization is an attempt to balance data quality and the work expended in the field and laboratory. The ultimate goal of was to develop a robust approach to measurement optimization, with the hope that a similar approach could be extended into other measurement network design problems."

Specific comments:

- in the title the root zone is explicitly mentioned however it doesn't appear later on to be an issue

We will change the title to

"Using Machine Learning to Predict Optimal Electromagnetic Induction Instrument Configurations for Characterizing the Shallow Subsurface"

- in the introduction you use the formulation "near surface hydrogeologic structure", later you switch to layered soils – maybe you can unify wording

We unified the wording to only use layered soils and changed the sentence to:

"Water movement through the vadose zone is often controlled by the near surface layering of soil."

- the introduction contains many information that are rather a methodological description of your work, e.g., line 57-58, 85-107, please address these issue in the methods chapter

We agree to move the description from l57-58 to section 3.1 and the initial explanations of machine learning (l85-107) to section 3.2. The following remains in the introduction to introduce the concept:

"Machine Learning (ML) describes a wide range of regression algorithms used for pattern recognition. ML has grown in popularity and is now used regularly within and beyond science. The simplest ML tools are based on Decision Trees (DT), which are supervised ML techniques that perform classification or regression by sequential categorization based on observations. DTs are computationally inexpensive, but they can have limited predictive skill (Hastie et al., 2001). To improve their performance, DTs are often augmented by ensemble learning methods such as bagging (Breiman, 1996) and boosting (Friedman, 2001)."

And the following is moved/added to method section 3.2

"We found that gradient boosting (Elith et al., 2008; Friedman, 2001) offered improved performance without adding unreasonable additional computational effort and it was used for all analyses. For our application, each modelled $EC_a$ value in the ensemble of the different EMI configuration represents a

feature in ML parlance. We then tested the ability of DT with GB to infer the correct value of each subsurface property given the $EC_a$ that would be measured with all the EMI configurations."

"We used the feature importance capabilities of DT with GB to identify which observed $EC_a$ values were most informative for the inference and eliminated all insensitive configurations. This allows us to find the optimal instrument configurations for each subsurface parameter without having to do inverse modelling. To examine the impact of independent knowledge of any of the subsurface properties, we then repeated this analysis for a subset of the soil models that met a given restriction, such as only those that had a thin upper layer or a high EC middle layer."

- In order to simplify your discussion and figures the height above ground could be released in a first step, since the assumption the all option are in any case available is misleading, e.g., I don't think its possible to carry an instrument with a coil distance of 4m at a height of 10 cm above ground along an agricultural or grassland transect. I completely understand that it is tempting to use all the information since ML is designed for big data, however for better understanding you could make use of Fig.2 in combination with some practical issues to reduce input heights.

The Department of Geoscience at Aarhus University has a Dualem421S system that can be towed behind an all-terrain vehicle (https://www.aarhusgeoinstruments.dk/dualem).  While most fields are not completely leveled, the towed instrument still secure uniform instrument height that is close to uniform.  In addition, there seems to be persistent interest in making measurements at multiple heights (e.g. ground placement and hip height) to improve information content.

- Do you have an idea why is the residuals in Fig 3 and 6 not evenly distributed? low EC values are overestimated and high EC values are underestimated - this aspect of heteroskedasticity needs to be discussed

We would argue that the skew is relatively small and limited to the extreme high and low values.  Most of the residuals are symmetric.  To explain the extreme values, we expect that this is due to the limits on the input values of 0 mS/m to 100 mS/m. Therefore, as the true cases approach this limit there are no EC values below the minimum (above the maximum) that can provide symmetric residuals.

- Fig. 4 I agree that a problematic condition for EMI is the thickness of a layer which is shown nicely for the thickness of A – the thickness of B should be even more challenging however this is not represented in the "outliers"

Fig 4. Shows the distribution of values within the outliers (1 std. off) from fig. 3. ECA is the parameter that is being inferred and the distribution of thickness A values in the outlier set show that small values of thickness A are dominant. While the value distribution of thickness of B is uniform and therefore no specific thickness of layer B makes a worse inference of ECA.

Fig 4. Could be reproduced for each of the five subsurface parameters, but we have chosen to only do it for ECA as is also the case with fig 3. and 6. This is partly because the same information is presented later (Fig. 7), but for all parameters instead.

Figure 2 of this document is Fig 7 from the manuscript. Here the center column represents the attempts to infer ECB. Changing the range of thicknesses of layer B is shown with gray markers. It is worth noting here that the modification of thickness B provides the most dramatic differences in NRMSE between the three

restriction patterns. With a high NRMSE for the thinning of the layer (triangle) and a low NRMSE for the thickening (square) of the layer. The high NRMSE for the thinning is larger when inferring ECB than for inferring ECA showing that a thin ECB is even more challenging to detect, as the reviewer surmised based on their experience.

[Figure]

*Figure 2 Figure 7 from the manuscript with a caption that reads: "The changes in inference of the five subsurface parameters (x-axis) are based on a comparison between the RMSE from restricted case divided by the range of the parameter (Y-axis). The lines show how well the parameters are predicted when all parameters are full range. The color shows which parameter that is being represented and the location and symbol represents the three restriction patterns skewed low (left nudged triangle), centered (centered dot), skewed high (right nudged square)."*

- the usage of an NRMSE is not clear to me if you intend to guide the user directly (l468)

We have edited the manuscript to make sure that the term, NRMSE, is clearly defined and to state simply that the use of NRMSE is to inform the user if the change in instrument setup will provide higher quality data (lower NRMSE) or lower quality data (higher NRMSE)

We added the following paragraph to the section 4.2 where NRMSE is first mentioned.

"The NRMSE of the parameter is a measure of how well the ML is able to infer the individual parameters and thus how estimable the parameters are. Because the ML is trained on EMI output the NRMSE also suggests how well the EMI instrument can detect the soil properties"

And also added the following at the specific line (468) you refer to:

"A low NRMSE will suggest a more reliable characterization of the subsurface property by the instrument and vice versa."

Referee references

McLachlan, P.J., Chambers, J.E., Uhlemann, S.S. & Binley, A. 2017. Geophysical characterisation of the groundwater–surface water interface. Advances in Water Resources, 109, 302-319.

Reyes, J., Wendroth, O., Matocha, C., Zhu, J., Ren, W. & Karathanasis, A.D. 2018. Reliably Mapping Clay Content Coregionalized with Electrical Conductivity. Soil Science Society of America Journal, 82, 578-592.

Robinet, J., von Hebel, C., Govers, G., van der Kruk, J., Minella, J.P.G., Schlesner, A., Ameijeiras-Mariño, Y. & Vanderborght, J. 2018. Spatial variability of soil water content and soil electrical conductivity across scales derived from Electromagnetic Induction and Time Domain Reflectometry. Geoderma, 314, 160-174.

von Hebel, C., Matveeva, M., Verweij, E., Rademske, P., Kaufmann, M.S., Brogi, C., Vereecken, H., Rascher, U. & van der Kruk, J. 2018. Understanding Soil and Plant Interaction by Combining Ground-Based Quantitative Electromagnetic Induction and Airborne Hyperspectral Data. Geophysical Research Letters, 45, 7571-7579.

Author references

Palacky, G. J. (2011). 3. Resistivity Characteristics of Geologic Targets. *Electromagnetic Methods in Applied Geophysics*, 52–129. https://doi.org/10.1190/1.9781560802631.ch3

---

## Author Comment (AC2)

**Response from the authors to the comments by anonymous referee**

We would like to thank the referee for providing constructive review and commentary.

**Dear Authors,**

Many thanks for your original contribution. Below you find my main comments and suggestions, which I hope will help you finalise your paper.

I have structured my comments per manuscript section, after my general remarks. References to specific parts of the text are made with line numbers (L.XX). In essence, I think the paper presents an interesting approach, but in certain aspects argumentation for specific hypotheses, decisions and conclusions is lacking or incomplete. This lack of (complete) argumentation for these aspects throughout the manuscript is why I have indicated the scientific significance of the manuscript in its current form as 'poor'. As such, I believe major revisions are required. In my opinion, these should address the general comments presented below.

**Thank you for finding the presented approach interesting.**

**GENERAL COMMENTS**

You propose a method to enable more efficient EM survey strategizing, mainly aimed at non-expert users.

You hereby start with the premise that the current means to determine the optimal EM instrument configuration, defined in your paper as a combination of coil configuration (geometry and spacing) and instrument height, are insufficient. However, it is unclear what you see as those current means (see for instance L.65-66 of the introduction)? It is therefore difficult to evaluate to which types of approaches you (want to) compare your approach. You equally do not define or specify the 'rules of thumb' for the application of EM instrumentation, again making it impossible to fully understand what you mean by this.

We now explicitly state the commonly applied rules of thumb within the introduction (see detailed comments).

Secondly, you start by stating that using modelling to predict the response of multiple soil models is computationally too challenging (I think that is what you mean in L.80). I don't think this is the case, particularly not for 1D modelling, as you perform yourself. So, either this point is incompletely made in the manuscript, or it may be (partially) incorrect. For one, simply presenting the sensitivities of the considered coil configurations would already elucidate much of their application potential.

During optimization of measurement campaign, it is required to consider the range of possible designs and the degree of uncertainty in the conditions being surveyed.

In the simplest case, a researcher may consider all but one property to be well defined and consequently only having one adjustable parameter. Furthermore, they may be choosing among several single instrument configurations. For these cases, a researcher can conduct a single-parameter sensitivity analysis and the coil configuration with the highest sensitivity can be selected.

Optimization becomes more difficult if the campaign includes the combination of multiple coil configurations. For these conditions, the shared information of all sensors must be considered. A researcher can still use a sensitivity analysis, but each combination of sensors must be considered, and it is not clear how the sensitivities of multiple coil configurations should be combined.

It is more appropriate to conduct an inverse analysis with each combination of sensors and to use this to infer the combined information in each measurement set. Further complication arises when more than one parameter value is unknown. For these conditions, it is no longer appropriate to conduct a single parameter sensitivity analysis assuming all but the parameter of interest are known. The number of sensitivity analyses (with an associated inverse model for each) increases geometrically.

Consider the following conditions. We can use up to 27 different coil configurations. There are five parameters that characterize the system to be surveyed. Each of these parameters are represented by a range of 10 different possible values. The forward model is extremely fast (0.01 seconds) and the inverse model is fast (1 second).

Considering a single sensor and a single parameter, with all others known, only 270 forward models need to be run for a total run time of 2.7 seconds.

If we consider a pair of coils, there are 351 combinations of coil configurations, each of which requires 10 forward models and 1 inverse model. This is already a total of 386 seconds.

In a study such as that shown here, we consider 27 sensors which may be taken 1, 2, 3, 4, or 5 at a time and all five parameters can vary. To determine the optimal set over this range of conditions would require consideration of  $10^5$  possible parameter combinations and (27 + 351 + 2925 + 17550 + 80730 = 101583) possible sensor sets.

Each parameter set requires one forward model (1000 seconds) and each sensor set requires one inverse model (10158 seconds). Therefore, taking a typical sensitivity analysis approach would require approximately 3 hours of simulation time.

In contrast, the approach described here requires the same number of forward models, the ML replaces the individual inverse models and only requires 60 min, and the underlying structure of an ML ensures that the solution balances goodness of fit with generalizability.

Then, if a user wants to know how information about one parameter might influence the survey design, they would down sample the forward models and, following a traditional approach as suggested, they would have to repeat all of the inverse models. That is, each examination of any set of existing information on the survey design would require hours of simulation time. In contrast, the ML only requires retraining on the reduced set of forward models which requires a few minutes.

It should be emphasized that the availability of EMagPy, and the nature of the EM problem, lead to very fast forward and inverse analyses. Many design problems require far more effort – especially if a more

complex forward model is chosen. The run time of the forward model is particularly limiting on traditional inverse analyses, which requires many forward models to be run during the inverse process. In contrast, the ML only requires that each forward model be run once, up front, and then the ML can be trained with a training time that is independent of the forward model run time.

You deploy machine learning to predict the optimal combination of coil configurations for targeting one of five subsurface parameters (EC + thickness of two layers, plus EC of a third layer with thickness set to infinity). Essentially, what you are doing is evaluating how sensitive the evaluated (27) instrument configurations are to each of these parameters. Or, more correctly, how sensitive the deployed forward model of those configurations is to these. Here, I do not fully see the difference between the machine learning approach you take towards this issue, and a simpler sensitivity analysis (e.g., Monte-Carlo based)? The latter, in my opinion, has at least two advantages: it is simpler (i.e., it is a straightforward, robust way to evaluate the influence of parameters on a model outcome), and it would be more straightforward to visualize.

As discussed above, the ML approach is fundamentally different – and we claim, more efficient – than a traditional inverse modeling approach to assessing data worth. First, an ML is trained specifically to balance goodness of fit with generalizability – this is not a common feature of inverse analyses. Second, the tree-based MLs used here conduct a data-worth analysis at each step as part of the training. This is imperfect, of course, but it naturally identifies data that do not contribute to the training and quantifies the contribution of the important inputs. In contrast, a sensitivity analysis (or, more accurately, and inverse analysis) would have to be repeated multiple times for subsets of the data to determine which combinations of observations are most informative. The key element of this work is the recognition that the feature importance provides a rank order of data worth that is produced without extra effort as the training seeks to improve the goodness of fit while avoiding overfitting. The result is that we can provide a very efficient calculation of the information content of different coil configurations for a user-defined range of site conditions. In this context, we think that a case can be made that this has clear advantages over a much more computationally expensive inverse analysis of data worth to support survey design.

Next, you deploy a forward modelling procedure, which you do not describe in detail. I think you use the (so-called) McNeil approximation, but you do not state this explicitly? You implement the modelling through EMagPy, but, again, without providing details on the model you use. This makes it difficult to evaluate the outcomes of your procedure (though I think you use McNeil, and have evaluated the following as such). If you use an approximation that is only valid within specific conditions (low induction condition), you essentially use a simplified (albeit elaborate) rule of thumb?

We appreciate the reviewer's point – we relied too heavily on the publication describing EMagPy. In the revised version we explicitly state that we use the McNeil approximation and provide a description so that the reader is not required to read the EMagPy paper.

"Then, the ECa was calculated for many EMI instrument configurations through EMagPy (Mclachlan et al., 2020) version 1.1.0. EMagPy deployed the CS response functions from eq. 1, 2 (McNeil, 1980), and 3 (Wait,

1962) in combination with the summation of eq. 4, which assumes that the LIN approximations (McNeil, 1980) are valid."

In your modelling procedure: you only consider quite a narrow range of EC variations (0-100 to meet – generally speaking – the LIN condition). This effectively limits the application potential of your approach (but would only imply deploying a forward model integrating the full solution – e.g. Hanssens et al. 2019

https://doi.org/10.1109/MGRS.2018.2881767). You equally do not consider other factors such as (instrumental) noise.

The EC range was chosen to both approximate the LIN condition and still cover the EC range of a large portion of agricultural fields. (Interestingly, the other reviewer commented that the EC range was too wide!) However, it should be noted that our approach is not reliant on using the forward model selected. A more complete forward model, such as that available in EMagPy, could be used or a user could even link to their own even more complete forward model, if it was necessary for their application. There would be no change to our approach other than the increased forward model run time. We chose the McNeil approximation because the model is widely used in the interpretation of EMI data and we believe that data collection and analysis should be linked. (That is, the data collection should be designed, to the degree possible, with consideration of the analyses that will be applied to the data.) Furthermore, we wanted to connect the results from ML with underlying concepts that would be accessible to a wide audience. We felt that this discussion would be easier to follow if we used the simpler McNeil solution rather than add considerations of changes in spatial sensitivity as a function of the EC structure.

The reviewer is correct that our analyses did not consider measurement noise. This would be an important extension of the work presented here, which is intended to be a proof of concept of a novel use of ML analysis for measurement network optimization.

**DETAILED COMMENTS PER SECTION**

**ABSTRACT**

L.14: There are general, rule-of-thumb guides to choose an optimal instrument configuration for a specific survey

While I understand this is not elaborated on in the abstract, you should explain which ones you mean and what the possible advantages/shortcomings are.

It is now explicitly stated which rules of thumb we refer to in the introduction (see detailed comments below).

L.15: The goal of this study was to use machine learning (ML) to improve this design optimization task

I assume the goal is to provide a robust, efficient way to strategize EM surveys. ML is not a goal, it is a tool.

The line is changed to:

"The goal of this study is to provide a robust and efficient way to design this optimization task."

**INTRODUCTION**

L.47: combined current is measured with a receiver coil

Magnetic field (cf. the following sentence)

The line now reads:

"combined magnetic field is measured with a receiver coil"

L.55: Finally, in some cases, the spatial sensitivity may depend on the absolute value and spatial distribution of the EC (Callegary et al., 2012).

What do you mean, in some cases?

The dependency of spatial sensitivity on the absolute value and spatial distribution of the EC are small when working at low frequency and low EC (LIN condition). The cases we refer to are the ones outside of the LIN conditions. Changed the wording to:

"Finally, in some cases, the spatial sensitivity may have a higher dependency on the absolute value and spatial distribution of the EC (Callegary et al., 2012)."

L.57: In this investigation, we make the common assumption that the spatial sensitivity only depends on the instrument configuration, but this dependence could be considered using more complete forward models of EMI response.

Why not use a fwd model that integrates the other relevant aspects (see general comments)

We kept this first situation simple to be able to connect the result of the machine learning analysis to the physical concepts and be able to present to the discussion to a wide audience (see general reply).

L.64: Developers of EMI instruments have long recommended using different configurations to measure layered ECa values, leading to simple rules of thumb such as

**using shorter coil separations for shallow mapping and larger separations for deeper investigations $\cdot$**

What are these rules of thumb you refer to? Make these explicit.

This is a good point. The rule of thumb we refer to is derived from the 70% cumulative response in LIN condition. Where 70% of the total response for VCP, HCP and PRP coils are accumulated from respectively 0.75, 1.5 and 0.5 coil separations of depths. In practice this leads to general rule of thumbs such as use a of short coil separation or VCP/HCP for shallow survey and larger separation or HCP for deeper survey. This is

We have made this more explicit in the introduction with the following:

"The depth of investigation (DOI) of EMI instruments is both in the scientific literature (Saey et al., 2009a; Saey et al., 2009b; Saey et al., 2012; De Smedt et al., 2014; Doolittle & Brevik, 2014; Adamchuk et al., 2015) and by the manufacturers (Dualem Inc., Canada n.d.) often estimated to be at the depth the has 70% of the cumulative response. There is a relationship between depth sensitivity of the instrument response and coil spacing and position. Therefore 70% cumulative response rule is in practice frequently converted to a rule of thumb that states larger coil spacings and HCP should be used for deeper investigations while short spacing and VCP/PRP should be used for shallow investigation (Acworth, 1999; Beamish, 2011; Cockx et al., 2009; K Heil & Schmidhalter, 2015; Kurt Heil & Schmidhalter, 2019). While this rule of thumb is not wrong the terms shallow and deep are subjective and will have different meaning depending on whether it is a hydrogeologist, archeologist, agronomist or a geophysicist who applies the terms. It also fails to make any distinction to the differences between using the VCP or PRP coil orientation."

**L.65: But little specific guidance is offered.**

What do you mean? I see:

- a/2 rule
- 70% cumulative response in LIN conditions (McNeill)
- Forward modelling

It is true that the 70% cumulative response rule provides general user guidance. It fails to provide specific information about what subsurface conditions would cause a short VCP coil to be better than a slightly longer PRP coil or an even shorter HCP coil.

Forward modelling will be able to calculate the instrument responses from different instrument configurations and subsurface layering it needs to be combined with another tool to rank the many combinations of configurations for different subsurface layering. Such as what we do with machine learning.

L.66: Furthermore, there is no way for a user to consider the possible impact of prior knowledge (e·g· bounds on the expected depth of the topmost layer) in the survey design.

Unless I am missing something here, this is not true. Forward modelling can easily provide this information.

Some kind of tool need to join the forward modelling procedure to analyze the vast number of possible outcomes. E.g. Monte Carlo (As you suggested) or ML that we deploy.

L.72: for t users

Users or the users.

**Corrected to "for users"**

L.72: This makes it difficult for users without theoretical background in geophysics to make an informed choice regarding the preferred instrument and configuration

This is a subjective statement: what do you mean by the theoretical background in geophysics required to deploy EM instruments? One could say that without the necessary understanding of basic theoretical concepts, you can never critically deploy this instrumentation. (I am aware that in practice this is not necessarily the case for all users)

It is our impression (based on personal experience) that a lot of users are guided by the rule of thumb that we describe, and it is very few that are aware of the underlying concepts, (apart from geophysicist).

We have changed the line to be:

"This makes it difficult to make an informed choice regarding the preferred instrument and configuration"

L.75: Each survey design includes multiple measurements at each location, each with a different configuration, that jointly provide the most useful information for inferring specific, user-identified subsurface properties.

survey or survey design?

**Survey design**

L.76: That is, a user is faced with the question of which combination of configurations is optimal given their measurement priorities and, ideally, incorporating any applicable constraints that they may have regarding the subsurface conditions. Any method that requires formal inversion of each proposed combination of configurations is computationally intractable for most users.

**What do you mean by this? Why inversion with survey design?**

**See general reply**

L.82 onwards: this is methodology, isn't it?

Yes, indeed, thanks for pointing this out. Majority of this section is moved to method section 3.2 that describes machine learning.

We kept some of the section within the introduction because we believe it is an important concept for this study.

"Machine Learning (ML) describes a wide range of regression algorithms used for pattern recognition. ML has grown in popularity and is now used regularly within and beyond science. The simplest ML tools are based on Decision Trees (DT), which are supervised ML techniques that perform classification or regression by sequential categorization based on observations. DTs are computationally inexpensive, but they can have limited predictive skill (Hastie et al., 2001). To improve their performance, DTs are often augmented by ensemble learning methods such as bagging (Breiman, 1996) and boosting (Friedman, 2001)."

L.77: Feature importance key ability of DTs (with and without GB), which is a functions that quantify the importance of each feature for making the predictions of interest.

**L.98: without having to do multiple inverse models**

**Forward models?**

Changed the sentence to: "This allows us to find the optimal instrument configurations for each subsurface parameter without having to do inverse modelling."

**THEORY**

L.113: It is more common, especially on agricultural soils

For (almost) all subsurface media

Changed line to "In almost all subsurface media the EC varies with depth due to soil layering."

L.119: low induction number

Explain (and reference)

The section has been changed to:

"The model only strictly applies under low induction number (LIN) conditions. The LIN approximation proposed by McNeil (1980) and assumes that changes in the measuring frequency has no effect on the response and that the depth of investigation does not depend on the EC of the subsurface. Assuming LIN conditions therefore means the response depends only on the depth, coil separation length, and coil configuration with no regard for the subsurface EC distribution."

L.120: with no regard for the subsurface EC distribution.

It would be good to mention that, generally, the output of commercially available EM instruments makes use of this approximation. The 'no regard for the subsurface EC distribution' is inherent to the ECa value, as you present yourself in the preceding paragraphs.

**Added the line**

"It is a common assumption for commercially available EMI instruments to operate under LIN conditions, despite being a simplification."

L.125: eq. 3: if I'm not mistaken, equation for PRP is based on Wait 1962, not McNeill (who only presents response functions for coplanar configurations)

Modified the beginning of the paragraph to include Wait 1962:

"The simplest, most widely used depth sensitivity model is the Cumulative Sensitivity (CS) model of McNeill (1980) (eq. 1 and 2) and Wait (1962) (eq. 3)."

L.136: EMagPy (McLachlan et al., 2020) offers the user the opportunity to use several models and makes them readily available to a wide audience, even users with no background in EMI modelling.

What do you mean by this? The fact that it incorporates a GUI?

Geophysical software licenses can easily be priced at several thousand euros. The fact that EMagPy is free makes it more available and appealing to someone who is not a specialist. Effectively increasing the number of people who can be participants in the geophysical community.

**METHODOLOGY**

L.150: using EMagPy

Ok, you mention the python package you use, but you should elucidate (and reference) the deployed forward model

Added a reference to equations 1, 2, 3 and 4 to the sentence:

"Then, the ECa was calculated for many EMI instrument configurations through EMagPy (Mclachlan et al., 2020) version 1.1.0, using the CS response functions from eq. 1, 2, and 3 in combination with the summation of eq. 4"

L.158: The lowest EC represents a dry sandy soil and the highest EC represent an agricultural soil with a combination of high clay, salinity, or water content

For a max. EC of 100, you cannot say that you evaluate the influence of salinity

Thank you for noting this we removed reference to salinity in the paragraph:

"The ranges of EC used in the forward model were chosen to represent a wide spectrum of soil types and water contents. The lowest EC represents a dry sandy soil and the highest EC represent an agricultural soil with a combination of high clay or water content (Triantafilis and Lesch, 2005; Robinson et al., 2008; Harvey and Morgan, 2009)."

L.169: from thin (0.05 m) to relatively thick (2.0 m) ...

Just state 'from 0.05 m to 2.0 m thickness'.

Changed the sentence to:

"The ranges of soil layer thicknesses span from 0.05 m to 2.0 m thickness."

L.164: Note that all analyses were repeated for the Andrade (2016) EMI model.

What do you mean by this? Explain the 'Andrade model'.

We removed the sentence at line 164:

"Note that all analyses were repeated for the Andrade (2016) EMI model "

L.165: The findings were not significantly different, so the results are presented for the simpler, more widely used McNeil model.

Because you stay within LIN conditions. I expect the difference to be most important (within LIN) for the PRP configurations?

Also, these are results, not methods

We removed the sentence at line 165:

"The findings were not significantly different, so the results are presented for the simpler, more widely used McNeil model."

L.179: x is the inputs (features) and 180 y is the response(

Rephrase

We modified the sentence to

"A training data set consists of n samples  $(x_1, y_1)$ ,  $(x_2, y_2)$ , ...,  $(x_n, y_n)$ , where  $x_{1-n}$  are the inputs (features) and  $y_{1-n}$  are the corresponding outputs (targets)."

L.198: gradient from which the algorithm named

Rephrase

We modified the sentence to:

"Right side of the minus sign in equation 6. is the gradient from which the algorithm is named, and the residual  $r_{im}$  are named pseudo-residuals."

L.221: optimal values for these parameters were found to be 0.1, 10, and 2, respectively

How?

We changed the sentence to be

"The learning rate, maximum tree depth, and minimum samples per leaf were tuned by manual trial and error and the optimal values for these parameters were found to be 0.1, 10, and 2, respectively."

L.229: Here, we examine how reducing the uncertainty of one soil EC parameter improves the EMI-based inference of other parameter values and whether this additional information changes the composition of the optimal EMI configurations to include in a survey.

Essentially a sensitivity analysis of your model/EM configuration to the EC and thickness of the respective soil layers you consider, which will be strongly related to the spatial sensitivity of the considered coil geometry.

Yes, this is a form of sensitivity analysis that looks at the boundaries for each parameter with the added bonus that it gives an estimate of the identifiability of the parameters given a specific soil.

**RESULTS & DISCUSSION**

L.255: The variations are less pronounced for larger coil separations.

as you would expect cf. spatial sensitivity of these geometries.

We agree that this is an expected outcome based on the spatial sensitivities.

L.256: differences in the smoothness of the distributions

I assume these are related to the EC of the upper soil layers? It is difficult to evaluate your results, as it is unclear which forward model you deploy. Is this just the 'McNeil-approximation'?

We added additional reference in the beginning of section 4 to make it more clear which forward models we deploy:

"In this section, we present the outcome from the forward modelling with the CS models for VCP (eq. 1), HCP (eq. 2) and PRP (eq. 3) and the summation from eq. 4 (section 4.1)."

L.301: The finding is opposite for ECA

if could it be the deployed forward model (approximation) strongly influences this as well? Furthermore, as this is (I think) still based on all 27 instrumeent configurations, this will have a significant influence as well. One would assume the poorly inferred cases are more likely related to configurations with a larger coil spacing?

These are the distribution of the parameter values from the outlier cases. Therefore, each case will indeed contain a response from each of the 27 configurations.

L.303: this suggests that the method would be more likely to be successful

Which method? Your approach?

Modified the sentence to:

"Practically, this suggests that identifying layer with an EMI instrument would be more likely to be successful"

L.305: A more successful survey, based on the ability to infer ECA, would occur if the ECA values tend to be lower. That is, a center or low skewed restriction should show better performance

Again: influence of the forward model?

The nature of the forward model will of course influence the outcome of the approach. But we want to verify the approach on a simple model before extending it to more complex forward models (see earlier comments).

**L.315: balances performance with reduced field effort**

What do you mean by this? You should clarify this aim in your introduction

Improving the quality of data or performance of the models that require said data while increasing the efficiency of field/lab work is an intrinsic part of optimizing experimental designs.

We have made this clearer in the introduction:

"One of the challenges of environmental investigations is to determine the optimal data to acquire. Data which is often used to provide structural information to a model or constrain model parameterization. Measurement optimization is an attempt to balance data quality and the work expended in the field and laboratory. The ultimate goal of was to develop an approach to measurement optimization that would be accessible to a wide range of users, with the hope that a similar approach could be developed for other measurement network design problems. The specific objective of this investigation was to present an approach to select sets of EMI configurations that are optimal given the specific survey goals and any independent knowledge of the subsurface electrical properties."

L.319: circle

Corrected the typo

L.323: However, he did not consider the PRP orientations.

Tabbagh (1986 – doi: https://doi.org/10.1111/j.1475-4754.1986.tb00386.x) did.

We are sorry we did not include that reference, very relevant (thanks). Reference to this is now included in section 4.3.

L.325: To our knowledge, no other method, short of exhaustive comparisons ofmany synthetic inverse analyses, would have been able to show that a single configuration was so clearly dominant for inferring ECC.

I disagree. Evaluating the QP sensitivity of a specific coil configuration to perturbing EC can be evaluated in a quite straightforward manner (see, for instance Hanssens et al. 2019 – doi: https://doi.org/10.1109/MGRS.2018.2881767 )

We were not aware of the work by Hanssens et al. 2019. Thank you for bringing this interesting study to our attention. They too use a "brute force method" of calculating the sensitivities with each forward model being resolved multiple times based on the number of layers. Conducting a global sensitivity analysis (using all the soil) would be exhaustive.

L.325: The small coil separation and low instrument height fit with general expectations, but the PRP orientation was not expected before conducting this analysis

Why not? And, conversely, why where you expecting the VCP/HCP to outperform PRP? Provide the full argumentation.

We did not expect VCP/HCP to necessarily outperform PRP but expected a more equal performance between the VCP and PRP sensor.

L.335: Perhaps more controversially, in the context of EMI instrument design and use, only 26% of the most informative configurations used the VCP orientation ...

Why is this controversial?

The EM38 sensors makes use of the VCP and HCP configurations and has a very widespread use. It is the most widely used EMI instrument in agriculture according to (Heil & Schmidhalter, 2017). It is counterintuitive that the most widely used instrument uses the least sensitive coil (VCP) rather than PRP. This also suggests that there is a gap between the community of EMI specialists and a large portion of end users.

L.339: This may be partially explained by the spatial sensitivities of the orientations

Why only partially? What you are doing is essentially evaluating the applicability of geometries/configurations with specific spatial sensitivities.

**Changed the sentence to:**

"This may be explained by the spatial sensitivities of the orientations"

L.341: high spatial sensitivity redundancy for the HCP and VCP

Why redundancy? You mean that these are not very complementary?

Based on the analysis we believe that the HCP/PRP pairing are more complementary relatively to the HCP/VCP pairing.

**section 4.4 Parameter restriction analyses.**

This may be a consequence of an incomplete understanding I may have on specific aspects of your ML (and your overall study aim), but I don't understand the point of this aspect. What will happen is that the uncertainty of the outcome will be reduced based on how sensitive your EM configuration (FWD model is) to a specific parameter. So, based on the previous section, you would expect that fixing the properties (EC and thickness) of the

first model layer (the most shallow layer) will have the strongest influence for most coil configurations.

This is essentially what you present in 4.4.2 (and emphasise in L.389: *The only clear exception was inferring ECA*, which showed a greater improvement by restricting *ThickA with a central or right skew*)

What we find is that reducing the range of thickness of A reduces the uncertainty of inferring ECA more than reducing the range of ECA itself. But fixing the properties of layer A is not necessarily the best option for inferring the remaining 4 parameters.

L.396 and beyond/ explanation for Fig. 7 'as a guide for planning an EMI survey':

I find this an overly complicated way to address the sensitivity of specific coil configurations to specific (combinations of) subsurface perturbations. I still do not see the advantage of your approach to a simpler sensitivity analysis.

We agree that Fig. 7 can be a bit of a handful. However, in practice it would not be needed to visualize all the combinations from Fig. 7. But rather put in the assumed ranges of each parameter from a targeted field and then compare it to the full range.

L.410: From the perspective of an experienced user of EMI surveys, most of these general conclusions will be obvious, which helps to confirm the validity of the proposed approach

This is an odd statement when put in the perspective of your study aims and introduction.

We want to ensure that the ML comes to reasonable conclusions. This is also why chose a simple forward model so we can verify the findings. If the approach was used on a model that describes a complex nonlinear system e.g. a groundwater aquifer then it would be difficult to confirm the result.

You mention there is '*no way for a user to consider the possible impact of prior knowledge*'. I think this is not true: you can use open-source forward models to do this. And I think you refer to this by stating that 'most of these general conclusions will be obvious'.

The 'no way for a user to consider the possible impact of prior knowledge' is in the introduction and is referencing to the rule of thumb (that we now explicitly define). The sentence has been altered and now makes a direct reference.

L.412: We see the value of this analysis as providing general guidance to less experienced users and to provide more fine-tuned guidance for site-specific conditions for those with more experience using EMI.

Essentially, you provide a means to evaluate different realisations of a forward model. This is indeed useful.

Thank you for finding this part of our study to be useful.

L.413: Furthermore, the guidiance provided is quantifiable rather than based on general rules-of-thumb.

You do not specify what you mean by 'rules of thumb'? You also do not compare the outcomes of your analysis to the assessments provided by these rules of thumb.

One could also consider using the 'McNeill-approximations' (i.e. approximations under the LIN condition) as a rule of thumb.

We now specify what we mean by rule of thumb in the introduction.

Our approach is not tied to any specific model or even domain of models. One of the advantages is that it is very general and can be extended to deal with any type of input and output. We just chose to showcase this with a simple geophysical model of EMI instrument response.

L.434: Figure & is somewhat information dense

Very true. Cf. my previous comments, a simpler sensitivity analysis would offer more clarity (and, I think, perhaps partially make section 4.4 redundant)

Each individual ring can be represented by a more traditional tornado diagram, but it would require a lot of diagrams to show all the combinations. This figure is designed to display all the states within the boundaries we defined. In practice a user could constrain the ranges and would display a pie for each parameter like in Fig 5.

L.444: This result could not be anticipated based on McNeil's solutions

What do you mean by this? Essentially, you are using the McNeil approximations, so I don't understand this statement?

Changed the sentence to:

"This result could not be anticipated based on the rule of thumb"

L.450: It is surprising, however, that one of the four observations place the instrument higher above ground  $\cdot$  We suggest that this is a good example of a result that has both

immediate practical value for survey design and could point researchers to ask follow-on questions about why this combination of observations is identified as optimal.

This is essentially a result of the spatial sensitivity (as captured in the deployed FWD model) of the evaluated configurations.

We believe the approach can be extended to other more complex forward models or combinations of forward models where the conclusions are less straightforward.

L.465: taken together Fig. 7 and 8 provide a direct guide to an EMI user when designing a survey with a specific target

Again, I think this is a very complicated guide. What you do in the section above is describe the observations you make in your analysis, based on the importance of features in your ML approach. You hereby circumvent discussing the physical basis for this, which lies in the spatial sensitivities of the EM configurations. Your discussion now is very descriptive and data-driven. While there is nothing wrong with this, essentially, I really think you cannot aim to provide practical insight into EM survey strategizing without laying out these fundamental theoretical concepts. This is, for instance, done very clearly by Tabbagh 1986 (see ref. above).

Our ultimate aim is to provide a robust method for design optimization partially by showing that ML is a reasonable approach, which is also one of the reasons we choose a simple model. So here, we show that it works for a simple geophysical model and we can make direct connections between the result of ML and expected outcome. If a user was in a situation where they could decide between multiple geophysical methods for their problem. They could essentially all be incorporated into the approach by extending the ensemble with appropriate forward models of higher complexity.

**References**

Heil, K., & Schmidhalter, U. (2017). The application of EM38: Determination of soil parameters, selection of soil sampling points and use in agriculture and archaeology. *Sensors (Switzerland)*, 17(11). https://doi.org/10.3390/s17112540

---

## Author Response (AR2)

**Response from the authors to the comments by editor Gerrit de Rooij**

We would like to thank the editor for taking the time to provide this feedback.

Dear authors,

I sent your revision back to the original reviewers and received one report.

Significant issues remain, but I concur with the reviewer that these are differences of opnion that can be addressed in scientific debate and should not disqualify the paper from publication.

But I do think you should be able to carry this debate over from the discussion (which will be published alongside the paper) to the paper itself. That is, I believe it is worthwhile to address the remaining issues in the paper.

The reviewer indicates some shortcomings in your approach that you can acknowledge without taking away from the main thrust of the paper (if you agree with them). There may be other cases where you disagree where it may well be possible to develop an counterargument in the discussion section or present and assess alternatives to your approach in the Introduction.

From the discussion so far it transpires that you seem to agree with many points raised in the review process, but did not use the revision to make modifications that reflect this. In effect, you are relying on the discussion to address points that can also be addressed directly in the main text.

I therefore ask you to go over the paper once more with this feedback in mind. Ideally, the text of the paper will reflect the outcome of the discussion that will accompany it. Since there were no new issues raised, I believe this should be achievable with minor revisions.

We thank the reviewers and editor for their constructive criticism.  We regret that it seemed that we were not open to revising the paper.  In fact, as detailed below, this version represents a significant change in the approach.  Namely, we are using the more complete forward model rather than the McNeil approximations.  As we had expected, the final results are essentially unchanged.  But, we agree with the author that using the more complete solution removes any concern about deficiencies of the widely used McNeil equations.  To highlight changes made in the text, we have included quotes from the revised paper in this response letter.

The argumentation in the discussion for using the simple model (McNeil) will not be carried over to the manuscript, because we have changed the forward geophysical model to a Maxwell-based full solution

Following added to section 2.1

"This study uses a complete forward model when estimating $EC_a$ to capture the changes in spatial sensitivity introduced by variation in subsurface EC. However, there is no hindrance to use a simpler geophysical model or a model describing a different process."

Comments from previous review that were not used in the revision to make a direct modification. Some of which are also addressed by changes from this round of replies.

Moreover the authors choose a rather arbitrary selection covering a very broad range of subsurface properties for the forward models. The chosen ECa range is rather high and from the practical point of view many field sites vary by a delta ECa not more than 20 mS/m which would cover only two classes (e.g., van Hebel 2018, McLachlan 2017, Robinet 2018, Reyes 2018).

"The ranges of EC used in the forward model were chosen to represent a wide spectrum of soil types and water contents. This is to capture different scenarios of EMI use e.g., a survey of a large heterogenous area. The lowest EC represents a dry sandy soil and the highest EC represent an agricultural soil with a combination of high clay or water content (Triantafilis and Lesch, 2005; Robinson et al., 2008; Harvey and Morgan, 2009). "

Given the option of EMagPy it seems to me more convenient, even for an unexperienced user, to run a forward model with several instrument configurations (HCP, VCP, PRP and coil distances) for the specific application with some prior knowledge of texture, salinity etc..

Added to section 3.1:

"Each of the three coil orientations was modelled for three different coil separations and three different instrument heights, the 27 instrument configurations cover both the more typical configurations for field applications of EMI and some more uncommon configuration."

Added to section 3.3

"This study uses layer EC and thickness as prior knowledge, but any information can be considered to constrain the range of cases."

Added to section 4.5

"The ML provides a quantitative measure of the shared information among model parameters (Table 2 and Fig. 7) to compare the likely success of each configuration."

Secondly, you start by stating that using modelling to predict the response of multiple soil models is computationally too challenging (I think that is what you mean in L.80). I don't think this is the case, particularly not for 1D modelling, as you perform yourself. So, either this point is incompletely made in the manuscript, or it may be (partially) incorrect. For one, simply presenting the sensitivities of the considered coil configurations would already elucidate much of their application potential.

Following

" In this study 27 instrument configurations in combination with 100000 subsurface models is considered the full ensemble. Using the presented ML approach to assess data value for our full ensemble is more

efficient than an inverse (Furman et al., 2007; Khodja et al., 2010; Song et al., 2016) or sensitivity (Hanssens et al., 2019) approach. In some cases, evaluating all instrument configuration will not be necessary, which means the inverse or sensitivity approaches become more efficient. The ML approach requires a certain size of model ensemble to yield stable results therefore model run time will reduce efficiency, but this affects the inverse analysis more because it generally requires more model runs. Ultimately the efficiency of the ML, inverse and sensitivity approaches depends, in the EMI case, on model run time, number of layers, parameter boundaries and the number of considered configuration and the combination of these in the applied case will determine which method is more efficient."

You equally do not consider other factors such as (instrumental) noise.

We now consider the effect of 5% gaussian measurement noise and added it as appendix B.

**Appendix B The effect of noise on Inferring subsurface parameters and feature importance**

To assess the impact of noise 100 realizations of heteroscedastic gaussian noise with a standard deviation of 0.05. The ensemble from the full solution was multiplied by the random noise prior to ML application to the full ensemble (no restrictions). This was repeated for each realization of noise and the average fit and their standard deviation are shown in table B1.

**Table B1 The root mean square error (RMSE) between the prediction from the gradient boosted (GB) model and the testing data. The machine learning procedure was repeated with each of the five subsurface parameters as targets, thus creating five models.**

| Target | ECA | ThickA | ECB | ThickB | ECC |
|---|---|---|---|---|---|
| Unit | mS/m | m | mS/m | m | mS/m |
| RMSE | 7.09 | 0.29 | 18.8 | 0.51 | 2.98 |
| RMSE (noise) | 12.4 | 0.41 | 23.3 | 0.58 | 8.00 |
| Std (noise) | 0.05 | 0.0008 | 0.05 | 0.0008 | 0.02 |

The average feature importance over the 100 realizations (Fig. B1) affects ThickA, ECB and ThickB the most. Here the feature importance is distributed more evenly among the configurations compared to without noise.

[Figure]

**Figure B1 Feature importance for inferring each of the five parameters from a decision tree analysis of the full parameter range. The feature importance from all 27 configurations sum to 1. The eight most important configurations for inferring each**

**of the five parameters are shown with a unique color and pattern combination. The remaining 19 configurations are aggregated into the "others" category and displayed with white.**

"

L.229: *Here, we examine how reducing the uncertainty of one soil EC parameter improves the EMI-based inference of other parameter values and whether this additional information changes the composition of the optimal EMI configurations to include in a survey*

Essentially a sensitivity analysis of your model/EM configuration to the EC and thickness of the respective soil layers you consider, which will be strongly related to the spatial sensitivity of the considered coil geometry.

Added to section 3.3

"Additionally, to the sensitivity of the configurations this analysis provides the parameter values that results in significantly lowered identifiability of any one of the five subsurface parameters"

L.465: *taken together Fig. 7 and 8 provide a direct guide to an EMI user when designing a survey with a specific target*

Again, I think this is a very complicated guide. What you do in the section above is describe the observations you make in your analysis, based on the importance of features in your ML approach. You hereby circumvent discussing the physical basis for this, which lies in the spatial sensitivities of the EM configurations. Your discussion now is very descriptive and data-driven. While there is nothing wrong with this, essentially, I really think you cannot aim to provide practical insight into EM survey strategizing without laying out these fundamental theoretical concepts. This is, for instance, done very clearly by Tabbagh 1986 (see ref. above).

Added to section 4.5

"Designing a combination of optimal configurations based on a conceptual understanding of the spatial sensitivities (rule of thumb) is not a reasonable task. Furthermore, measurement optimization requires a quantitative measure of the information content. The ML provides a quantitative measure of the shared information among model parameters (Table 2 and Fig. 7) to compare the likely success of each configuration."

**Response from the authors to the comments by anonymous referee**

We would like to thank the referee for taking the time to provide this constructive review.

Dear authors,
many thanks for the detailed responses. I gather from your responses that we disagree on a couple of aspects regarding your methodology and its meaning, but that of course is not a problem. It makes things interesting.

Dear Referee, thank you for finding the discussion interesting.

My main general concerns remain similar to those in the previous review round and are related primarily to [1] the use of an incomplete forward model, and [2] the usefulness of possible alternative approaches to yours by conducting a more basic sensitivity analysis. I stand by my previous review comments that these factors should be addressed. The limitations of your study (which does lean strongly on aspects of EM theory) should be properly stated.

For [1] I do think you should mention (or, ideally do) the following:
- assessment of the impact of an incomplete forward model on a selection of results. Ideally, you would use a more complete forward model to evaluate the limitations of this aspect of your methodology. This is raised by other reviewers, but you ignore this. Either state this very clearly or - ideally - provide a fuller solution approach as a comparison. The fact that 'McNeill is still the most widely used ...' is not a valid argument. It is incomplete and it would serve the use of EM methods better if alternative open-source models (again, as presented in the previous review round) are available.
I strongly disagree with: ' Furthermore, the connection between results from ML analysis and theory becomes simpler and the discussion is accessible to a wider audience.'. How does something become simpler when it has fundamental limitations that strongly impact the interpretative potential of your results, and you do not mention these impacts? You also state that there is no hindrance to use a more complete forward model. If that is the case, why have you not done this?

We agree that it serves EM usage and the community to use more complete forward models if open-source options are available. Both to achieve more realistic modelling results and spread the knowledge of the software that offers the complete solutions.

We have therefore repeated all our modelling (EM and ML) and results with a Maxwell-based full solution that EMagPy offers. These results (tables and figures) replace the previous ones where we used the simple CS model (rather than accompany), because we want the focus to be on the suggested approach itself. For our cases the outcome did change slightly, but the overall picture remained.

Therefore, the theory section is now:

"We apply the Maxwell-based full solutions (eq. 1, 2 and 3) from Wait (1982) to calculate the relationship Q between the secondary field ($H_s$) and the primary field ($H_p$). The solution works for a one-dimensional subsurface and it is valid for low frequencies because it assumes that the electromagnetic fields spread due to conduction currents:

$$Q_{VCP} = Im\left(\frac{H_S}{H_P}\right)_{VCP} = Im\left(-s^2 \int_0^\infty R_0 J_1(s\lambda)\lambda d\lambda\right), \tag{1}$$

$$Q_{HCP} = Im\left(\frac{H_S}{H_P}\right)_{HCP} = Im\left(-s^3 \int_0^\infty R_0 J_0(s\lambda)\lambda^2 d\lambda\right), \tag{2}$$

$$Q_{PRP} = Im\left(\frac{H_S}{H_P}\right)_{PRP} = Im\left(-s^3 \int_0^\infty R_0 J_1(s\lambda)\lambda^2 d\lambda\right), \tag{3}$$

Where Im means that only the imaginary component is considered, $R_0$ is an interlayer reflection factor, $J_0$ and $J_1$ are Bessel functions of respectively zeroth and first orders and λ is the radial wave number. The integrals of eq. 1, 2 and 3 represent Hankel transform and in the EMagPy software (McLachlan et al., 2020) these are calculated with linear filtering (Anderson, 1979; Guptasarma & Singh, 1997). The LIN approximation proposed by McNeil (1980) assumes that depth of investigation does not depend on the EC of the subsurface. Therefore, a method similar to that of von Hebel et al. (2019) is used through EMagPy (McLachlan et al., 2020) to estimate $EC_a$ from Q. The $EC_a$ is estimated by minimizing the differences between a predicted or measured $Q_{pred}$ and a Q value calculated for an equal homogenous half-space, $Q_{homo}$. The minimized difference approach is valid for a broader range of $EC_a$ compared to the LIN approximation (Von Hebel et al., 2019; McLachlan et al., 2020). We refer to Von Hebel et al. (2019) for a more detailed description of this method."

To the left are the previous McNeil based figures and tables and to the right is the full solution based figures. The differences are mainly revealed through the feature importance when inferring the ECA, ECB and ECC (fig 5 and 8).

**Table 2: The root mean square error (RMSE) between the prediction from the gradient boosted (GB) model and the testing data. The machine learning procedure was repeated with each of the five subsurface parameters as targets, thus creating five models. The RMSE is normalized by the mean value of the target to get the normalized root mean square error (NRMSE).**

| | McNeil | | | | | Full solution | | | | |
|---|---|---|---|---|---|---|---|---|---|---|
| Target | ECA | ThickA | ECB | ThickB | ECC | Target | ECA | ThickA | ECB | ThickB | ECC |
| Unit | mS/m | m | mS/m | m | mS/m | Unit | mS/m | m | mS/m | m | mS/m |
| RMSE | 7.34 | 0.29 | 18.7 | 0.49 | 1.51 | RMSE | 7.09 | 0.29 | 18.8 | 0.51 | 2.98 |
| NRMSE | 0.07 | 0.20 | 0.19 | 0.26 | 0.02 | NRMSE | 0.07 | 0.20 | 0.19 | 0.27 | 0.03 |

[Figure]

**Figure 1: The result from running the DT with GB on the entire 100000 soil types and all 27 instrument configurations five times. The EC of the A-layer (ECA) is the parameter that is being predicted. The X-axis is the true value of the ECA, and the Y-axis is the predicted values for ECA.**

[Figure]

**Figure 2: The ECA was inferred for 150,000 test cases. In 8894/8816 of the 150,000 cases the inference was more than one standard deviation away from the true value. The figure shows the distribution of five subsurface parameter values within the 8894 conditions. The top X-axis is the layer thickness, the bottom X-axis is the layer EC and the Y-axis is the frequency.**

[Figure]

**Figure 3: Feature importance for inferring each of the five parameters from a decision tree analysis of the full parameter range. The feature importance from all 27 configurations sum to 1. The eight most important configurations for inferring each of the five parameters are shown with a unique color and pattern combination. The remaining 19 configurations are aggregated into the "others" category and displayed with white.**

[Figure]

**Figure 4: The result from running the machine learning algorithm on a subset of the ensemble where the thickness of the A-layer have been restricted. Only 20,000 soil types and all 27 instrument configurations remain in this restricted subset. The EC of the A-layer (ECA) is the parameter that is being predicted.**

[Figure]

**Figure 5: The changes in inference of the five subsurface parameters (X-axis) are based on a comparison between the RMSE from restricted case divided by the range of the parameter (Y-axis). The lines show how well the parameters are predicted when all parameters are full range. The color shows which parameter that is being represented and the location and symbol represents the three restriction patterns skewed low (left nudged triangle), centered (centered dot), skewed high (right nudged square).**

[Figure]

**Figure 6: Feature importance for the 8 most important EMI configurations for every combination of the five inferred/restricted parameters and the three patterns. Each circle is subdivided into four rings that shows, from inside out, the feature importance for full range, centered, skew low, and skew high. Each column/row represents the each of the five inferred/restricted parameters. The coil orientations are colored so that Horizontal (HCP) is blue, Vertical (VCP) is green, and Perpendicular (PRP) is red. A dark and light hue represents respectively a short and long coil distance.**

For [2] i do think that in many cases specific knowledge on the survey area etc. is available or can be gathered. Uncertainties (which is, I assume, what you in your responses dub as bias) can be integrated in such procedures to a certain extent (and are always inherent to geophysical applications/modelling). The argument that conducting an analysis that includes all possible EM instruments is only partly valid. Usually (always?) users evaluate a limited set of available instruments/configurations so, while absolutely useful and key in your manuscript, the evaluating the full suite of instruments is – in my opinion – not a standard requirement.

We have added the following to the discussion (section 4.5)

" In this study 27 instrument configurations in combination with 100000 subsurface models is considered the full ensemble. Using the presented ML approach to assess data value for our full ensemble is more efficient than an inverse (Furman et al., 2007; Khodja et al., 2010; Song et al., 2016) or sensitivity (Hanssens et al., 2019) approach. In some cases, evaluating all instrument configuration will not be necessary, which means the inverse or sensitivity approaches become more efficient. The ML approach requires a certain size of model ensemble to yield stable results therefore model run time will reduce efficiency, but this affects the inverse analysis more because it generally requires more model runs. Ultimately the efficiency of the ML, inverse and sensitivity approaches depends, in the EMI case, on model run time, number of layers, parameter boundaries and the number of considered configuration and the combination of these in the applied case will determine which method is more efficient. Designing a combination of optimal configurations based on a conceptual understanding of the spatial sensitivities (rule of thumb) is not a reasonable task. Furthermore, measurement optimization requires a quantitative measure of the information content. The ML provides a quantitative measure of the shared information among model parameters (Table 2 and Fig. 7) to compare the likely success of each configuration"

Furthermore (dotting the I's), you do not fully integrate the instrument specifications (e.g. frequency and, more importantly, noise level). As an argument you state that you present a proof of concept of a novel use of ML analysis for measurement network optimization. That is only partly true, for your main manuscript, the balance between focus on EM and ML is about 50/50, so I believe you cannot use this argument as a rationale for using simplifications and omitting factors such as noise. At the very least, you should make this rationale – and the associated limitations – explicit in your manuscript.

We assessed the effect of noise. These results are added to appendix B in the manuscript. With a reference in section 4.2

"a table of how gaussian noise affect the RMSE is shown in appendix B."

And section 4.3

"The influence of simulated noise on the results in Fig. 5 are shown in appendix B."

**"Appendix B The effect of noise on Inferring subsurface parameters and feature importance**

To assess the impact of noise 100 realizations of heteroscedastic gaussian noise with a standard deviation of 0.05. The ensemble from the full solution was multiplied by the random noise prior to ML application to the full ensemble (no restrictions). This was repeated for each realization of noise and the average fit and their standard deviation are shown in table B1.

**Table B1 The root mean square error (RMSE) between the prediction from the gradient boosted (GB) model and the testing data. The machine learning procedure was repeated with each of the five subsurface parameters as targets, thus creating five models.**

| Target | ECA | ThickA | ECB | ThickB | ECC |
|---|---|---|---|---|---|
| Unit | mS/m | m | mS/m | m | mS/m |
| RMSE | 7.09 | 0.29 | 18.8 | 0.51 | 2.98 |
| RMSE (noise) | 12.4 | 0.41 | 23.3 | 0.58 | 8.00 |
| Std (noise) | 0.05 | 0.0008 | 0.05 | 0.0008 | 0.02 |

The average feature importance over the 100 realizations (Fig. B1) affects ThickA, ECB and ThickB the most. Here the feature importance is distributed more evenly among the configurations compared to without noise.

[Figure]

**Figure B1** Feature importance for inferring each of the five parameters from a decision tree analysis of the full parameter range. The feature importance from all 27 configurations sum to 1. The eight most important configurations for inferring each of the five parameters are shown with a unique color and pattern combination. The remaining 19 configurations are aggregated into the "others" category and displayed with white.

"

I hope this helps finalize the manuscript.

It did indeed, thanks again for the feedback.

---

## Author Response (AR3)

**Response from the authors to technical corrections made by editor Gerrit de Rooij**

The authors would like to thank the editor for taking the time to provide these corrections.

**Comments to the author:**

Dear authors,

I think the latest round made the paper substantially more solid, both in the content and in the way you argue your case. I am therefore happy to inform you that I can accept the paper for publication.

Thank you for accepting the paper for publication. We agree that the latest round strengthened the paper.

I only have a few formatting details that will come up anyway as the paper is processed, and that should only take a few minues to take care of:

lines 133, 137, 138, 159, 160 (perhaps more): symbols in regular font in the text appear in italics in the equations. Please convert the text fonts to italics as well.

We have edited the in-text symbols in section 2.1 and 3.2 to be italic like the equations they refer to.

And changed units in figures 2, 3 4 and 6 to be written exponentially.

l. 134: Hankel transform -> Hankel transforms

Edited

The caption of Fig. 7: there are no x- or y- coordinates in the figure. Perhaps replace 'x-axis' and 'y-axis' by horizontal 'axis' and 'vertical axis'.

Changed the wording in Fig. 7 caption from x- and y-axis to horizontal- and vertical axis respectively.